# Experimental Dust Absorption Study in Automotive Engine Inlet Air Filter Materials

**DOI:** 10.3390/ma17133249

**Published:** 2024-07-02

**Authors:** Tadeusz Dziubak

**Affiliations:** Faculty of Mechanical Engineering, Institute of Vehicles and Transport, Military University of Technology, Gen. Sylwestra Kaliskiego Street 2, 00-908 Warsaw, Poland; tadeusz.dziubak@wat.edu.pl; Tel.: +48-261-837-121

**Keywords:** air filter materials, dust absorption, composite materials, separation efficiency and accuracy, pressure drop, vehicle mileage

## Abstract

The purpose of this study was to empirically evaluate the performance of fibrous materials that meet the criteria for inlet air filtration in internal combustion engines. The characteristics of filtration efficiency and accuracy, as well as the characteristics of flow resistance, were determined based on the mass of dust accumulated in the filter bed during the filtration process. Single-layer filter materials tested included cellulose, polyester, and glass microfiber. Multilayer filter media such as cellulose–polyester–nanofibers and cellulose–polyester were also examined. A new composite filter bed—consisting of polyester, glass microfiber, and cellulose—and its filtration characteristics were evaluated. Utilizing specific air filtration quality factors, it was demonstrated that the composite is characterized by high pre-filtration efficiency (99.98%), a short pre-filtration period (*q_s_* = 4.21%), high accuracy (*d_pmax_* = 1.5–3 µm) for the entire lifespan of the filter, and a 60–250% higher dust absorption coefficient compared to the other tested materials. A filtration composite bed constructed from a group of materials with different filtration parameters can be, due to its high filtration efficiency, accuracy, and dust absorption, an excellent filter material for engine intake air. The composite’s filtration parameters will depend on the type of filter layers and their order relative to the aerosol flow. This paper presents a methodology for the selection and testing of various filter materials.

## 1. Introduction

Road surfaces on which motor vehicles travel serve as collection points for pollutants originating from both natural and artificial sources. The primary component of these pollutants is mineral dust, which is carried by the wind from the surrounding soils—especially those disturbed by fieldwork and road construction activities—as well as from vast sandy areas and deserts. In addition, the road surface can collect dry tree and plant leaves, pollen, animal dander, mold, and other contaminants of biological origin [1,2]. These materials can be transported from nearby areas and from along the roadside. In addition, dust is generated during various agricultural and industrial processes, including the harvesting, baling, milling, drying, and compacting of processed herbaceous biomass, as well as during the mechanical industrial processes of processing agricultural crops such as grain and cotton [3].

A considerable portion of dust pollution is attributed to human activities, including industrial production, mining operations, and motorization [4]. In addition, contaminants on road surfaces often come from abrasion processes in brake and clutch friction pairs [5,6] and from wear on the road surface and tire tread. These particles, once released into the atmosphere, settle back down to the ground due to gravity [7,8,9,10].

Pollution also includes substances such as soot and abrasive wear products of the engine’s metal components, which are expelled from the engine through the exhaust pipe. In addition, road surfaces may be covered with grains of salt used to destroy the ice layer. The dirt with which the road surface is covered can be lifted upward by vehicles traveling at high speed or by gusts of wind, taking the form of dust. This phenomenon is commonly referred to as road dust [11,12,13]. Its main components are the minerals silica (SiO_2_), having a content of 65–95%, and aluminum oxide Al_2_O_3_ (4–18%). The remainder are oxides: F_2_O_3_, CaO, MgO and organic components [14].

The combustion process in an engine requires a minimum of 14.5 kg of air to burn 1 kg of fuel, whether gasoline or diesel. Internal combustion engines of passenger cars draw about 140–450 m^3^/h of ambient air, while engines of special cars and trucks consume about 900–1400 m^3^/h. The engines of tracked special vehicles require more than 6000 m^3^/h of air for operation. Vehicle internal combustion engines suck in significant amounts of pollutants from the environment, and mainly mineral dust, which, due to its content of hard silica and aluminum oxide grains, is the main cause of accelerated wear of frictionally cooperating engine components.

Studies have shown that an engine’s lifespan is highly dependent on the cleanliness of the intake air [15,16,17,18]. It has been shown that wear of engine components is mainly caused by particles of 1–40 µm, but particles in the 5–20 µm range are the most dangerous [19,20,21,22,23]. In addition, surface wear is determined by the hardness of the dust grains. Silica (SiO_2_) and alumina (Al_2_O_3_) grains, the total content of which reaches 95% in dust, have a hardness of 7 and 9, respectively, on the ten-grade Mohs scale [24]. Thus, they are superior in hardness to the structural materials used for internal combustion engine components (cylinder liners, crankshaft journals and camshafts). In addition, mineral dust grains have very irregular shapes and sharp edges. The most dangerous for the two mating engine components are dust particles whose diameter *d_p_* is equal to the thickness of the oil film *h_min_* between the two surfaces at any given time.

Therefore, the hardness of these dust grains exceeds that of the structural materials used in internal combustion engines, such as cylinder liners, crankshaft journals, and camshafts. Moreover, mineral dust grains are characterized by their very irregular shapes and sharp edges, contributing to their abrasive properties.

The value of dust concentration in the air varies within wide limits and depends on many factors: terrain conditions, type of ground (sandy or paved), traffic conditions (traffic volume and speed, movement of other vehicles, driving alone or in a column, and exhaust emissions), meteorological conditions (type of precipitation and frequency, wind direction and strength), road surface condition (dry, wet, ice-covered, salted, or sand-covered), vehicle design and running gear (wheeled, tracked), and the location and design of the engine air intake [25,26,27]. Table 1 shows examples of the dust concentration values in ambient air [28].

A study [9] showed that, under actual operating conditions, the mass of a vehicle and the intensity of braking significantly affect the emission of PM2.5 and PM10 particles from brake and tire wear. In the study, the highest concentrations of PM2.5 (520–4280 μg/m^3^) and PM10 (950–8420 μg/m^3^) particles from brake wear were observed for the vehicle with the highest mass, while the lowest peak values of PM2.5 (250–2440 μg/m^3^) and PM10 (430–3890 μg/m^3^) were observed for the vehicle with the lowest mass. Similarly, the highest values of PM2.5 (340–4750 μg/m^3^) and PM10 (810–8290 μg/m^3^) came from the tires of the heaviest vehicles. As expected, lighter-weight vehicles emitted much lower values of PM2.5 (340–4750 μg/m^3^) and PM10 (810–8290 μg/m^3^) from tire wear.

Estimates indicate that about 30% of the dust that enters the engine cylinders with the air may pass through the exhaust system unchanged, thus contributing to increased particulate matter (PM) emissions from the engine. It is estimated that only about 10–20% of the dust that enters the engine with the air through the intake system is retained on the walls of the cylinder liner. This retained dust then comes into contact with the piston and piston rings, causing abrasive wear on these components. Dust grains having a melting point much lower than the temperature generated in the combustion chamber (about 2000 °C) melt and enter the exhaust system with the flue gasses in the form of droplets. They can deposit on the walls of the catalytic reactor in the form of a glassy layer, which deteriorates its operation [29].

Excessive wear of the piston rings and cylinder liner enlarges the clearance within the piston–piston ring–cylinder (P-R-C) system. This enlargement is the main reason for the increased flow of compressed cargo into the engine crankcase. Such a scenario leads to reduced pressure at the end of the compression stroke, engine torque and horsepower, and increased fuel consumption, as well as increased exhaust emissions [30,31,32,33,34]. Consequently, this wear and tear negatively impact engine life and vehicle reliability, leading to more frequent maintenance requirements and potentially higher operational costs [35].

Utilizing air filters in engine intake systems is the only effective strategy to shield engines from inhaling harmful environmental pollutants. These pollutants significantly accelerate the wear on engine components and drastically impair engine performance. Air filters vary in design, operating principles, and the materials used for filtration. Modern passenger car engines, which predominantly operate on paved roads where the air dust concentration is relatively low, are outfitted with single-stage filters. These filters feature a cartridge composed of pleated filter material. Conversely, trucks, work machinery, and tracked special vehicles, which are commonly operated in environments with high dust concentrations in the air, benefit from employing two-stage filters. This setup comprises an inert pre-filter and a main filter, the latter also using a cartridge made of pleated filter material, arranged in series to provide enhanced protection against dust and particulate matter [36].

The engine intake air filter exerts both positive and negative effects on engine performance. Positively, it significantly mitigates abrasive wear on engine components and prevents a reduction in engine power by filtering out harmful particulates. However, integrating an air filter into the engine intake system introduces a pressure drop, which translates into additional energy costs and can lead to a decrease in engine torque and power [37,38]. In modern passenger cars, the space around the engine is limited due to the need to install new equipment. As a result, it reduces the summation of an air filter with a large enough area of filter material, which further aggravates the issue of pressure drop. This necessitates the use of highly absorbent materials to counteract this effect.

The primary challenge lies in balancing the need for a low pressure drop and extended filter life with the requirement for high filtration efficiency and accuracy in removing dust particles. Composite filter materials present a promising solution to this dilemma. They offer the potential for achieving a favorable balance between maintaining engine performance and protecting the engine from particulate-induced wear, all while ensuring the filter’s longevity and effectiveness.

## 2. Literature Analysis of Filtration Properties of Composite Deposits

The primary filter media used for purifying the intake air of modern internal combustion engines, indoor air quality, and the air inhaled by humans through masks predominantly consist of cellulose-based fibrous materials [39,40].

These cellulose-type filter media are known for their high durability relative to some alternative materials. They are composed of relatively large fibers, typically with diameters exceeding 10–15 µm. As a result, while they offer significant advantages in terms of strength and longevity, their separation efficiency and filtration accuracy might not always meet the highest standards, particularly during the initial period of filtration. The filtration process of contaminated air in fibrous materials is characterized by the occurrence of an initial period, which is characterized by low efficiency *φ_f_* and filtration accuracy *d_pmax_*, as well as a low pressure drop Δ*p_f_* at the beginning of the process [41].

However, as the mass of dust retained on the filter bed increases, its performance parameters: efficiency *φ_f_*, filtration accuracy dpmax and flow resistance Δ*p_f_* change, taking on increasing values [41]. The initial filtration efficiency of the paper bed has an insignificant value and is *φ_f0_*, = 96.3%, but as a result of the accumulation of dust in the bed, the filtration efficiency increases steadily. The assumed (target) value of *φ_f_*, = 99.5% filter is achieved at the dust absorption coefficient *k_d_* = 110.7 g/m^2^ and the presence of maximum dust grains of *d_pmax_* = 10–28 µm in the air. These are the grains that are the cause of accelerated wear of machine components.

To meet the stringent requirements for engine inlet air filtration, which include high separation efficiency (above 99.5%) and filtration accuracy for particles larger than 2–5 µm, along with extended service intervals, cellulosic materials often undergo various modifications. One of the significant advances in improving separation efficiency and filtration accuracy in fiber beds is the use of nanofibers made of polymers with diameters less than 1 µm. The diameters of standard cellulose fibers are in the range of 15–20 µm. A thin layer of nanofibers most often having a thickness of 1–5 µm is applied to a substrate of conventional filter materials, which are characterized by greater thickness and strength. The nanofibers can be deposited on one or both sides of the substrate, which can consist of materials such as cellulose, nylon, glass microfiber or polyester.

Incorporating nanofibers as an additional layer on standard air filter materials significantly boosts both separation efficiency and accuracy. By applying a thin nanofiber layer on the inlet side of a standard filter bed, such as cellulose, dirt particles are trapped before they can penetrate deeper into the filter material. This approach has been extensively researched and developed in both academic and industrial settings, underscoring the potential of nanotechnology in improving air filtration systems for motor vehicles and other applications, thereby enhancing performance and prolonging the lifespan of engine components [42,43,44,45,46,47,48,49,50,51,52].

The authors of the paper [42] proposed a bed having a micro-nano layered structure and investigated its application in air filters. The composite filter consists of an outer layer of polystyrene (PS) microfibers with high electrical resistance and an inner layer of polyacrylonitrile (PAN) nanofibers with high polarity and small pore size. It was found that the PS/PAN/PS composite filter had a high separation efficiency of 99.96% for 0.30 μm particles, a low pressure drops of 54 Pa and a satisfactory quality factor value of *q_c_* = 0.1449 L/Pa at an air flow velocity of 0.053 m/s.

The authors of the paper [43] proposed a composite material obtained from microcellulose fibers made of cellulose pulp reinforced with bacterial nanocellulose. The study showed that the composite bed has properties that allow it to achieve high (above 99.5% for PM1 and PM2.5) filtration efficiency and low pressure drop of 0.194 kPa/g.

The authors of the paper [44] used a one-step co-spinning method to fabricate and then test a three-layer nanofiber composite bed (PET/TPU-CNF). The bed design had a low flow resistance of 28.9 Pa and a NaCl particle filtration efficiency of 83.6% at a filtration velocity of 0.053 m/s. The mechanical and filtration properties of the PET/TPU-CNF bed proved to be superior compared to a composite filter made of PET nanofibers only.

The results of testing the filtration efficiency of four samples made of different filter materials are presented in the article [45]. New nanofibers composites were used on four different fabric structures. The filtration efficiency of material samples without a nanofiber layer is low and for dust grains smaller than 2 μm does not exceed 10%. Applying a thin layer of nanofibers (*g_m_* = 0.02 g/m^2^) to the filter bed of the tested samples increases the separation efficiency of particles below 2 μm to more than 60%.

The paper [46] presents the fractional performance of a filter medium based on cellulose nanofibers, on the surface of which a layer of nanofibers was applied with the following parameters: thickness *g_z_* = 0.3 mm, grammage *g_m_* = 0.1 g/m^2^ and fiber diameter *d_fb_* = 40–800 nm. The filtration efficiency of the designed bed was determined using dust with a grain size of less than 10 μm. The filtration efficiency of the developed bed in the range *φ* = 64–99% was obtained by setting the filtration velocity *υ_F_* = 0.03 m/s and dosing dust grains in the range *d_p_* = 0.2–4.5 μm. Increasing (almost tenfold) the filtration speed to *υ_F_* = 0.2 m/s reduces the filtration efficiency significantly.

The authors of the paper [47], using nano-CaCO_3_ powder as a test dust, determined the filtration properties of the PTFE membrane-coated filter medium. Compared to conventional filter material, the separation efficiency of the PTFE membrane-coated filter medium was greater than 99.99% for micron particles.

The study of the influence of the content of nanofibers (share: 5%, 10%, 15%, 20%) in the cellulose filter bed on the filtration efficiency was presented in [48]. An increase in the share of nanofibers in the filter bed results in an increase in filtration efficiency and a simultaneous increase in flow resistance. A 10% share of nanofibers means a threefold increase in flow resistance and filtration efficiency of dust grains with a size of *d_p_* = 0.8 µm. Increase in size Dust grains in the range of *d_p_* = 0.03–0.2 µm cause a decrease in filtration efficiency, and in the range of *d_p_* = 0.2–2 µm it increases. The effectiveness reaches its lowest value for particle size around 200 nm, regardless of the percentage of nanofibers in the bed. The efficiency curve is shifted almost parallel to higher values if the nanofiber compactness in the bed is increased.

In the paper [49], a new sub-microfiber filter medium was presented to provide dust protection for engines. The new medium was prepared by wet jointing two filter layers with a paper machine. During laboratory and field tests, the dust efficiency of the new filter bed was compared with other similar filter media. The mass dust absorption capacity of the new filter bed was higher than that of the standard filter bed and the submicrofiber bed, by 48% and 10%, respectively. Flow resistance in the sub-microfiber filter was found to be 45% lower compared to the standard Heavy-Duty filter after field testing over 10,000 km.

The main aim of the research presented in [50] was to determine the influence of the pore size of the fibrous filter bed and the fiber diameter on its filtration efficiency. The tests used three types of standard polyester filter beds and two polyester filter beds covered with a polytetrafluoroethylene membrane (surface filtration) with different structure parameters (pore size and fiber diameter). It was found that the greater the increase in filtration efficiency and fabric flow resistance, the smaller the pore size of the filter bed.

The authors of the paper [51] presented tests of a filter composed of two filter elements arranged in series (preliminary filter material and main material) placed close to each other. A low-efficiency polypropylene synthetic filter material was used as the initial material. The pre-filter has a large-diameter fiber backing, and thus has a high air permeability of 3100 L/(m^2^/s). Therefore, it has low efficiency and accuracy for filtering small dust grains. The main filter is a PTFE membrane material, which consists of a coarse cellulose substrate with a laminated PTFE membrane. Thus, it has low permeability and high efficiency for removing fine particles. To reduce the filtration speed, the filter material was pleated. A significant reduction in the intensity of the increase in pressure drop of the main filter and an increase in its service life were achieved, as well as an increase in the total particle retention capacity of the entire filtration system.

Studies of the properties and separation efficiency of three types of PTFE membranes are presented in [52]. Tubular PTFE membrane filters of different diameters, lengths and membrane layers were prepared and tested for their ability to remove particulate matter. Tests were performed for particles from 10 to 300 nm at face velocities in the range of 0.003–0.15 m/s. Membranes with thicker and smaller pores had higher performance, and the performance curves for PTFE sheet membranes showed a typical “V” shape from 10 to 300 nm at face velocities from 0.003 to 0.15 m/s. According to the test results, the filter containing two layers of membranes had the lowest pressure loss while achieving a very high particle removal efficiency of more than 99.98% for particles (0.3 µm) and almost 100% for particles (2.5 µm).

The analysis provided yields several key insights into the performance of filter beds used for engine intake air filtration:filter beds with a layered structure demonstrate superior separation efficiency, accuracy, and dust absorption capabilities compared to single layers or different materials combined into a multilayer bed. This suggests that the strategic arrangement and composition of layers significantly enhance filtration performance.although there is research on multilayer beds, their direct application in the context of removing contaminants from engine intake air faces limitations due to specific operating conditions such as air dustiness and dust particle sizes. It underscores the need for further research into filter deposits, taking these unique conditions into consideration to optimize filtration efficiency.the filter materials from which the filter beds of automobile intake air filters are manufactured usually have the structural parameters of the material (such as pore size, air permeability and thickness) specified in the product certificate. However, there is a lack of detailed information on their dust filtration properties, especially the unit absorption rate. This gap highlights the need for more comprehensive data to better evaluate and compare the performance of filter materials under realistic conditions.

The exploration into the efficacy of multilayer beds for the filtration of intake air in motor vehicle engines underscores a significant opportunity to enhance vehicle performance. Given that multilayer beds exhibit a higher dust absorption capacity compared to single-layer configurations, they have the potential to extend the operational life of vehicles by allowing them to operate longer before reaching the limit of permissible resistance. However, the key to unlocking this potential lies in validating whether such configurations can also meet the requisite standards for separation efficiency and accuracy.

The accumulation of dust within the filter bed, while beneficial for trapping pollutants, inevitably leads to an increase in pressure drop across the filter. An overly rapid increase in pressure drop could prematurely reach the threshold of permissible resistance, necessitating earlier than expected filter maintenance. This dynamic between dust accumulation, pressure drop, and filtration efficiency is critical yet not extensively covered in the existing literature.

To address these gaps and answer pertinent questions, a study was designed using a pleated filter bed comprising three base materials—polyester, glass macrofiber, and cellulose—each chosen for their distinct properties and known structural parameters. The objective of this research was to validate the potential of multilayer beds to improve separation efficiency and accuracy, alongside their dust-absorbing capacity. By achieving these improvements, the study aimed to increase the operating time of the engine inlet air filter, thereby reducing wear on engine components and extending their service life. Experimental tests conducted on both single layers and a newly developed multilayer bed evaluated their filtration properties and flow resistance in relation to unit dust absorption *k_d_*. While testing multilayer beds for inlet air filtration is recognized as both costly and labor-intensive, it is considered the most reliable method for obtaining accurate insights into the performance of filtration systems. Such research is invaluable for advancing our understanding of how best to protect engine components from airborne pollutants, ultimately leading to more durable and efficient engine designs.

## 3. Materials Used and Methods Employed

### 3.1. Study Subject

Tests were given to cylindrical filter cartridges (Table 2) made of pleated material, having the same filtration area *A_w_* = 0.183 sq.

Each cartridge was made of a different filter material, which were conventionally designated A (cellulose), B (glass microfiber), C (cellulose–polyester–nanofiber), D (cellulose–polyester), E (high-quality polyester) and composite K (polyester–glass microfiber–cellulose). The filter beds (A, B, C, D, E) are factory-made products and the composite bed K was made for the study by folding the three filter layers E, B, A at the pleating stage—Figure 1. Table 2 shows the characteristic parameters and selected properties of the filter materials provided for testing.

To facilitate the analysis of the test results, the filter materials (cartridges) are conventionally designated A (cellulose), B (glass microfiber), C (cellulose–polyester–nanofibers), D (cellulose–polyester), E (polyester) and composite K (polyester–glass microfiber–cellulose). SEM images (inlet and outlet sides) of the samples before testing are shown in Table 3.

Observations for assessing the microscopic stereometry of the surface of the samples, as well as for image registration, were conducted using a Philips XL30 scanning electron microscope (SEM, Philips XL30 TMP, F.E.I. Company, Hillsboro, OR, USA) equipped with an X-ray spectrometer and an “EDAX” attachment (Philips XL30 TMP, F.E.I. Company, Hillsboro, OR, USA). The Philips XL30 FEG SEM utilizes a Schottky field-emission gun design that features a point-source cathode made of tungsten, coated with a surface layer of zirconia (ZrO_2_). The operational temperature of the emitter is maintained at 1800 K. The inclusion of an X-ray spectrometer and an EDAX attachment further enhances the microscope’s capabilities, enabling elemental analysis alongside the topographical and morphological characterization of the sample surfaces.

### 3.2. Methodology and Test Conditions

The research was conducted to determine and compare the filtration properties: separation efficiency, filtration accuracy, and pressure drop of filters made of different filter materials (Table 2) by determining their characteristics:filtration accuracy *d_pmax_ = f*(*k_d_*),separation efficiency *φ_f_ = f*(*k_d_*),pressure drop Δ*p_f_ = f*(*k_d_*).

And flow (aerodynamic) characteristics Δ*p_f_ = f*(*Q_f_*), where *Q_f_* is the air flow through the filter bed, and *k_d_* is the dust absorption coefficient, determining the total mass of dust *m_z_* retained and uniformly distributed over 1 m^2^ of the active surface of the filter material, which is expressed by the following relation:(1)$$k_{d}=\frac{m_{F}}{A_{w}}\;\left[g/m^{2}\right]$$

The filtration velocity was defined as the quotient of the air flow through the filter cartridge *Q_f_* and the active area of the filter paper *A_w_* according to the relationship:(2)$$\upsilon_{F}=\frac{Q_{f}}{A_{w}\times 3600}\;\left[m/s\right]$$

The test stand (Figure 2) made it possible to record, using a particle counter, the number and size of dust grains in the air *Q_f_* behind the tested filter. Particles in the range of 0.7–100 µm were recorded, distributed in 32 measurement channels defined by fixed diameters (*d_pimin_* − *d_pimax_*).

In the process of evaluating a filter, the measuring probe’s tip is strategically positioned at an appropriate distance behind the filter being tested. It is centrally aligned with the axis of the conduit through which air is drawn into the particle counter’s sensor. The measuring tube is concluded with a specialized filter designed to block dust from entering the rotameter, ensuring the accuracy of flow rate measurements.

The test dust is uniformly introduced into the dust chamber, and its chemical and fractional composition is shown in Figure 3. This dust mixes with the air flow within the chamber, creating a two-phase gas-solid flow conducive for testing. The PTC-D test dust is formulated to be chemically and fractionally equivalent to Arizona fine test dust, a benchmark commonly employed in the assessment of intake air filters for motor vehicle and work machinery engines. 40% of the total weight of the test dust are dust grains less than 5 µm in size. These very small grains pose a significant challenge for retention by porous filter materials, due to their size.

A substantial portion, more than 67%, of these fine grains are composed of SiO_2_, a mineral known for its high hardness level of 7 on the Mohs scale, which spans from 1 to 10. The high hardness of SiO_2_ grains contributes to the accelerated wear of engine components, underlining the critical need for effective filtration to safeguard engine integrity and extend its lifespan.

The flow characteristics Δ*p_f_ = f*(*Q_f_*) of the filter cartridges were determined for 10 measurement points in the air flow range *Q_f_* = *Q_fmin_* − *Q_fmax_*. The maximum flux value *Q_fmax_* was determined for the assumed maximum filtration velocity *υ_F_ =* 0.1 m/s. For passenger car filters, the maximum filtration velocity of papers ranges from *υ_Fw_* = 0.07–0.1 m/s [25,27,41,47]. For the filtration velocity (*υ_F_* = 0.1 m/s), the maximum value of the test flux calculated according to the following relation has the value *Q_fmax_* = 65 m^3^/h.
(3)$$Q_{fmax}=A_{w}\cdot v_{F}\cdot 3600\;\left[m^{3}/h\right]$$

The test flow rate *Q_f_* was determined using an FMT430 mass flow meter with a measuring range of 10–150 m^3^/h and an accuracy of 1.2%. The basic characteristics of the filters (A, B, C, D, E): efficiency *φ_f_ = f*(*k_d_*) and filtration accuracy d *d_pmax_ = f*(*k_d_*), as well as flow resistance Δ*p_f_ = f*(*k_d_*) was made on the basis of parameters measured simultaneously by the same methodology on the test bench (Figure 2). The tests were performed at a constant air flow rate of *Q_f_* = 56 m^3^/h. This corresponds to a filtration velocity of *υ_F_ =* 0.085 m/s in the filter material.

To determine the parameters of the filtration process (efficiency and accuracy, as well as flow resistance), the mass method was used by repeating successively measurement cycles of the same duration *τ_pd_* and for the same measurement conditions. The filtration efficiency *φ_f_ = f*(*k_d_*)was determined using the gravimetric method in successively repeated “*j”* measurement cycles of constant duration *τ_pd_*. Before the start of the measurement cycle, the mass of the dust tank, the filter under test, the mass of the safety (absolute) filter was determined. For this purpose, an analytical balance with the following parameters was used: measuring range 220 g, accuracy 0.0001 g. The filters were fixed in the housing, and the dust tank in the metering device. After establishing the value of air flow *Q_f_* = 56 m^3^/h, dust was dosed uniformly into the tested filter at *τ_pd_*. The measurement time was set to *τ_pd_* = 2 min. when the tests were implemented during the initial period of the filtration process and *τ_pd_* = 4–5 min. during the tests of the head period of the filtration process.

The procedure for measuring the number and size of dust grains in the air stream behind the filter was started at the particle counter 60 s before the scheduled end of the measurement. During one measurement cycle, three particle counts (every 8 s) were programmed at the scheduled measurement intervals (*d_pimin_* − *d_pimax_*). After the measurement (dust dosing) was completed, the Δ*h_m_* value of the pressure drop behind the test filter was read, and the airflow was closed. After the test filter and absolute filter bed were removed from the housing and the dust container was removed from the dispenser, their weights were determined with an analytical balance.

The mass of dust *m_Dj_* that was lost from the dust container of the metering unit during the measurement and the mass of dust *m_Fj_* that was retained on the filter under test during this time, as well as the mass *m_Aj_* retained on the absolute filter, were determined as the difference in masses of the dust container and filters before and after each measurement cycle. The mass of the filter under test and the mass of the dust tank and the mass of the absolute filter after the measurement are also the masses before the next measurement.

After each measurement cycle *j* was determined as follows:filter efficiency from the following relation:
(4)$$\varphi_{fj}=\frac{m_{Fj}}{m_{Dj}}=\frac{m_{Fj2}-m_{Fj1}}{m_{Dj1}-m_{Dj2}}\cdot 100\%,$$
where 1 and 2 stand for the mass before and after the measurement of the dust tank and test filter, respectively.

2.Filter flow resistance Δ*p_fj_* of the filter was determined as the difference in static pressures upstream and downstream of the filter based on the measured height ∆*h_mj_* on a U-tube water manometer. This was measured at a fixed air flow rate *Q_j_* after the dust dosing was completed. The relationship used for this was:

(5)$$∆p_{fj}=\frac{∆h_{mj}}{1000}\cdot \left(\rho_{m}-\rho_{H}\right)\cdot g\;\left[\mathrm{Pa}\right]$$
where *ρ_m_*—density of manometric fluid [kg/m^3^]; *ρ_H_*—density of air [kg/m^3^]; and *g*—acceleration due to gravity [m/s^2^].

3.The dust mass loading *k_dj_* of the filter material in the filter from the formula:

(6)$$k_{dj}=\frac{\sum_{j=1}^{n}m_{Fj}}{A_{w}}\;\left[g/m^{2}\right]$$
where, $\sum_{j=1}^{n}m_{Fj}$—the total mass of dust retained on the test filter after successive measurement cycles.

4.The number *N_pi_* of dust particles in the purified air stream *Q_f_* (number of dust particles passed through the filter) in the designated intervals, which were limited by the conventional diameters (*d_pimin_* − *d_pimax_*).5.Filtration accuracy, which was defined as the largest dust grain size *d_pj_* = *d_pmax_* that was recorded in the air stream *Q_f_* downstream of the filter.6.The proportion of each dust grain fraction (given as a percentage) in the treated air (downstream of the filter) for each measurement cycle:

(7)$$U_{pi}=\frac{N_{pi}}{N_{p}}=\frac{N_{pi}}{\sum_{i=1}^{32}N_{pi}}100\%$$
where $N_{p}=\overset{32}{\underset{i=1}{\sum }}N_{pi}$ is number of dust grains in the exhaust stream *Q_f_* (from all measuring compartments) during the test cycle.

During the tests, a measurement cycle was used in which three counts of dust grains in the air behind the filter were programmed from the range 0.7–80 μm, which was divided into 32 measurement intervals limited by diameters (*d_pmin_ − d_pmax_*).

According to the methodology presented, the characteristics of the filters were determined: efficiency *φ_f_ = f*(*k_d_*) and filtration accuracy *d_pmax_ = f*(*k_d_*) and pressure drop Δ*p_f_ = f*(*k_d_*) of filter cartridges A, B, C, D, E and K. Two copies of each filter cartridge were tested with the same filter material and under the same conditions. The flow characteristics of the cartridges Δ*p_f_ = f*(*Q_f_*) was carried out before and after the test dust tests of the cartridges.

## 4. Test Results and Their Analysis

The results of testing the flow characteristics Δ*p_f_* = *f*(*Q_f_*) of filter cartridges made of different filter materials are shown in Figure 4. As the air flow rate increases, there is a parabolic increase in the pressure drop Δ*p_f_* = *f*(*Q_f_*), which is due to the increase in flow velocity in the quadrature and is consistent with the information found in the literature.

For the K filter, where the filter bed is a composite of three sequentially arranged layers: polyester, glass microfiber and cellulose (E + B + A), the highest values of flow resistance were recorded over the entire range of air flow *Q_f_* = 10–65 m^3^/h. For *Q_f_* = 56 m^3^/h, the pressure drop of the K cartridge has a value of Δ*p_f_* = 0.902 kPa (Figure 4). This value is more than 60% higher than the pressure drop for filter A with cellulose material. Such significant values of pressure drop are mainly due to the thickness of the composite bed, which consists of layers of three materials.

The pressure drop of the other filter materials is at a much lower level in the range Δ*p_f_ =* 0.511–0.735 kPa for a flux of *Q_f_* = 56 m^3^/h.

The results of testing the filtration characteristics of the efficiency *φ_f_ = f*(*k_d_*) and filtration accuracy *d_pmax_ = f*(*k_d_*) and the pressure drop Δ*p_f_ = f*(*k_d_*) of the factory-made filter materials (A, B, C, D, E) and the composite K made by the author for the purpose of this study are presented in Figure 5, Figure 6, Figure 7, Figure 8, Figure 9, Figure 10, Figure 11, Figure 12, Figure 13, Figure 14 and Figure 15. Two filter media were tested for each filter material.

The obtained characteristics of the filter cartridges (Figure 5, Figure 6, Figure 7, Figure 8, Figure 9, Figure 10, Figure 11, Figure 12, Figure 13, Figure 14 and Figure 15) are similar as to the waveform, but depending on the type of filter medium from which they are made, they differ as to the values. Dust flowing with the air stream is retained with the participation of filtration mechanisms in the filter layer and accumulated inside it. An increase in the mass of dust accumulated in the filter layer corresponds to an increase in the dust absorption coefficient *k_d_*. In addition, there is an increase in filtration efficiency and accuracy, as well as the flow resistance of the filter media. This is characteristic of fibrous deposits, except that the increase in filtration parameters during the initial period is more intense and varies in values for each cartridge. This is due to the different properties (parameters) of the filter materials, such as permeability, bed weight and pore size (Table 2).

With further increase in the *k_d_* coefficient, the filtering efficiency of the cartridge reaches maximum values (close to 100%), after which the efficiency usually remains at this level. The size of the maximum dust grains *d_pmax_* initially has large values, after which it decreases and remains constant, and at the end of the filtration process it usually takes on increasingly larger values. The pressure drop, with the increase in the dust mass on the filter cartridge, increases its value all the time, with the intensity of this increase being much higher in the final stage of filtration. It was assumed that the criterion for the termination of the filter cartridge tests was that the cartridge reached a fixed resistance value, called the permissible resistance Δ*p_fdop_*, provided that there was no rapid decrease in separation efficiency beforehand. An acceptable resistance value of Δ*p_fdop_* = 4 kPa was assumed for the filter materials tested.

The changes in the values of filter parameters demonstrated during the tests result from changes that occur in the structure of the filter bed as a result of retaining dust grains on the elements of the fibrous bed. This happens as a result of the main filtration mechanisms: inertial, direct attachment and diffusion. As a result of their action, incoming dust particles are retained and deposited on the surface of the fibers, where they form a layer. Further dust grains settle on it, creating subsequent layers. Consequently, large clusters of dust grains are formed, filling the free spaces (pores) between adjacent fibers.

Dust agglomerates forming on the fibers cause changes in the air flow conditions due to the decreasing distances between subsequent elements, which causes an increase in the flow speed. The consequence of this is an increase in hydrodynamic resistance in the filtration layer. At the same time, smaller distances between the fibers with the dust layer increase the intensity of the filtration mechanisms, which increases the filtration efficiency. This parameter reaches higher and higher values and approaches 100%, then stabilizes.

Figure 5 and Figure 6 show the efficiency *φ_f_ = f*(*k_d_*) and filtration accuracy *d_pmax_ = f*(*k_d_*) and the flow resistance Δ*p_f_ = f*(*k_d_*) of the filter cartridges, respectively: A (cellulose) and B (glass microfiber).

For cartridges having the same type of filter material, the differences in the course of characteristics and the obtained parameter values are insignificant. Analyzing the obtained characteristics of the filters, it was noted, a certain regularity in that the filtration process of each filter can be divided conventionally into two stages taking as a criterion the changes and values of filtration efficiency. The initial stage (*t_p_*) lasts from the moment the dust arrives on the filter until the filtration efficiency reaches the set value of *φ_f_ =* 99.9% (Figure 5). This is the filtration efficiency required for materials applicable to the construction of intake air filters for motor vehicle engines. This level of efficiency is a guarantee of minimal wear on those engine components that ensure its high durability and reliability. The next stage of the filtration process has been called the main stage (*t_g_*) and lasts until the filter reaches the flow resistance value set by the manufacturer, which is the criterion for ending the filter’s operation. This involves replacing the contaminated filter element with a new one. This operation is necessary because the pressure drop across the filter bed increases with the amount of accumulated dust particles, resulting in a reduction in air flow rate to the engine. This results in energy losses to the engine (decreased power and increased fuel consumption) and increased vehicle operating costs. In some air filter solutions, instead of replacing the contaminated cartridge, a pulsed cleaning operation is used [54,55]. The method involves removing dust particles collected by the filter bed during the filtration process, using a stream of compressed air delivered as a pulse to the cartridge in the direction opposite to the flow of the air stream during operation. This procedure allows continuous operation of the baffle filter. It follows that to ensure long engine life and low energy losses, the operation of the air filter should occur from the moment the filtration efficiency *φ_f_* = 99.9% is achieved (the end of the initial stage *t_p_*) until the permissible flow resistance is reached. This period (main stage *t_g_*) should be as long as possible, and the initial period, i.e., the time to achieve filtration efficiency *φ_f_* = 99.9% should be minimal.

A comparative analysis of the obtained filtration parameters of materials A and B is shown in Figure 7 based on the characteristics of filter No. 1.

The efficiency of the filter with material A takes an initial value of *φ_f0A_* = 96.3%, while for the filter with material B, the filtration efficiency takes a value at a much higher level of *φ_fA_* = 99.88%. This may be due to the lower permeability (*q_p_* = 190 dm^3^/m^2^/s) of material B and smaller pore size (*d_p_* = 35 µm) than material A (Table 2). Thus, the time to achieve the established filtration efficiency *φ_fA_* = 99.9% (the duration of the initial period *t_p_*) for cartridge A is much longer. This value is achieved by filter A at *k_dpA_* =14.9 g/m^2^, and cartridge B already at *k_dB_* = 7.45 g/m^2^, which represents 22.3% and 7.99%, respectively, of the total cartridge operating time limited by the achievement of the permissible flow resistance Δ*p_fdop_* = 4 kPa. For this value of flow resistance, materials A and B achieve a mass dust load of *k_dA_* = 66.9 g/m^2^ and *k_dB_* = 93.2 g/m^2^.

The evolution of pressure drop during the dust loading process (Figure 5 and Figure 6) shows an initial slow increase followed by a rapid increase. The general trend in flow resistance changes is similar to fiber filters under dust loading reported by other researchers [56]. The authors of the paper [56,57] divided the dust accumulation process of a fibrous test filter into three stages: the initial stage—a slow linear increase (I) of about 0.28 Pa per 1 g/m^2^ loaded dust mass, the transition stage—a rapid non-linear increase (II) and the final stage—an intense linear increase (III) at a rate of 3.85 Pa per 1 g/m^2^ loaded dust mass. The permissible flow resistance of 2.5 kPa was reached by the filter after the dust load on the bed reached 140.7 g/m^2^.

For filter A tested in this study, the slow initial growth rate is about 21.7 Pa per 1 g/m^2^ of loaded dust mass, and in the final stage the growth rate is 70.8 Pa per 1 g/m^2^ of loaded dust mass. For filter B, the intensity of the increase in flow resistance is much lower and is respectively: 8.13 Pa per 1 g/m^2^ and 62.18 Pa per 1 g/m^2^ for the final stage of the filter.

For the purpose of this work, it was assumed that the quotient of the values of the two coefficients of the mass dust load of the same filter material “*F*” expressed in (%) is the filtration efficiency index *q_s_* and is expressed by the following relation:(8)$$q_{s}=\frac{k_{dpF}}{k_{dF}}\;100\%$$
where *k_dpF_* is the coefficient of mass loading of the filter bed with dust after the end of the initial period *t_p_*, and *k_dF_* is the coefficient of mass loading of the filter bed with dust when the filter reaches the permissible flow resistance Δ*p_fdop_* = 4 kPa.

The qs coefficient determines the percentage of the initial period in the total filtration period determined by the permissible resistance, which corresponds to the dust absorption coefficient *k_dF_*. If two filter materials have the same *k_dF_* value, the one with the lower *k_dp_* coefficient value has better filtration properties and will achieve the established filtration efficiency earlier. The duration of the initial period of the filtration process will be shorter.

When the filtration process begins, there are dust grains of different diameters in the air behind the filter. The dust grain with the largest size indicates the accuracy of filtration. In the air behind cartridge A (cellulose), the size of dust with the largest size *d_pmax_* = 12 µm was recorded. Behind cartridge B, a much smaller size was recorded (*d_pmax_* = 7 µm). A close relationship can be seen between the value of initial efficiency and filtration accuracy (Figure 7). For filter A, the initial efficiency is 96.3% and the corresponding accuracy is 12 µm. For filter B, the values are 99.88% and 7 µm, respectively. They thus form slowly growing complex dendritic structures (agglomerates), which fill the free spaces between the fibers. The result of the changes in the bed structure is that the efficiency of the filtration mechanisms increases, and smaller and smaller dust grains are found in the air behind the filter. At the end of the first filtration period, dust grain sizes are at *d_pmax_* = 3–4 µm for both cartridges. The duration *t_p_* of the initial stage of the cartridges is particularly important for the service life of equipment requiring high-purity air, such as vehicle internal combustion engines. High air separation efficiency and accuracy is particularly important when supplying air to car drivers’ and passengers’ cabins. Low separation efficiency and large dust grain sizes (above 5 µm) in the intake air have a significant impact on the accelerated wear of engine components, mainly the piston, piston rings and cylinder liner. This situation occurs after the contaminated filter element has been replaced with a new one, when the initial period of filter operation begins.

The essential working period of cartridges A and B (Figure 7) is characterized by high (at 99.95%) separation efficiency, small grain sizes (*d_pmax_* = 3–4 µm), but a continuous increase in pressure drop. The permissible pressure drop value of Δ*p_fdop_* = 4 kPa was reached by cartridge A at a dust absorption coefficient of *k_dA_* = 66.9 g/m^2^, and cartridge B at a much higher value of *k_dB_* = 95.7 g/m^2^. Once the filter cartridges reach Δ*p_fdop_* = 4 kPa, increasingly larger dpmax grain sizes are recorded in the exhaust air. For Δ*p_fdop_* = 8 kPa, the grain sizes take on a value of *d_pmax_* = 8 µm, with no sharp drop in separation efficiency recorded. The significantly higher dust-absorbing capacity of filter material B (micro glass fiber) compared to filter A, is due to the much greater (double) thickness of the material B bed (*g_zA_* = 0.395 mm, *g_zB_* = 0.76 mm—Table 1), which allows it to collect a greater mass of dust.

From the test results shown in Figure 5 and Figure 6, it can be seen that there is a close relationship between the efficiency of the filter and its flow resistance regardless of what material they were made of. The characteristics of filters made of different materials have a similar course, but they differ not only in the value of the parameter, but also in the fact that at the end of the initial period they reached the established filtration efficiency *φ_f_* = 99.9% at different dust absorption coefficients. Another feature that differentiates the tested filters is the value of the kd coefficient obtained at the assumed permissible flow resistance. Direct comparison of the filtration properties of different materials is difficult. Therefore, the literature commonly uses the filtration quality factor qc, which refers to the separation efficiency and pressure drop of the same filter material. According to [58,59], the filtration quality factor is expressed by the following relation:(9)$$q_{c}=\frac{-\ln\left(1-\varphi_{0}\right)}{{∆}_{p}}\;\left[{\mathrm{kPa}}^{-1}\right]$$
where *φ*_0_ is the initial separation efficiency, and Δ*p_f_* is the pressure drop.

The higher the value of the *q_c_* coefficient, the more effective the filtration process. From the values of the filtration quality coefficient *q_c_* and the separation efficiency coefficient *q_s_* of materials A and B presented in Figure 8, it can be seen that the filtration properties of material A (cellulose) differ significantly from material B (glass macrofiber).

This is mainly expressed in the long (more than 30%) initial filtration period *t_p_*, which is characterized by low separation efficiency and accuracy. It follows from the above that the cellulose bed, due to its low efficiency and low accuracy, is not a suitable material for filtration of the air sucked in by the vehicle engine. The use of such a filter may significantly accelerate engine wear, which results in a decrease in power and increased emissions of toxic gasses in the exhaust gasses.

For this reason, other fiber compounds such as polyester and a layer of nanofibers are added to the cellulose bed.

Figure 9, Figure 10 and Figure 11 show the characteristics of filter materials that are a blend of different materials: C (80% cellulose, 20% polyester–nanofibers) and D (cellulose–polyester) and high-grade polyester E.

A comparative analysis of the characteristics *φ_f_ = f*(*k_d_*) and *d_pmax_* = *f*(*k_d_*) and Δ*p_f_ = f*(*k_d_*) depending on the dust absorption coefficient kd of filters: C, D and E is presented in Figure 12.

A summary of the filtration characteristics for the three filter materials labeled C, D, and E illustrates (Figure 12) that, akin to the results observed with materials A and B (Figure 5 and Figure 6), the performance metrics of these filter materials exhibit similar trends but vary in magnitude depending on the specific filter material composition. With an increase in the mass of dust retained in the filter layer (an increase in the *k_d_* coefficient), the separation efficiency and accuracy as well as the pressure drop of the filter media tested take on increasing values. It can be seen directly from the figure that filter material C (cellulose–polyester–nanofibers) obtained the highest initial efficiency of 99.8%, the shortest initial filtration period, but also the lowest value of the absorption coefficient *k_d_*. The good filtration properties obtained by this material are due to the nanofibers layer used, which results in small dust grains in the range of 2–4 µm in the cleaned air during the main working period. The high separation efficiency means that the filter material collects larger dust masses, hence the intensity of the increase in pressure drop is greater than that of other materials. This leads to an earlier achievement of the permissible pressure drop Δ*p_fdop_* = 4 kPa. This is confirmed by the values of the filtration quality coefficients *q_c_* and separation efficiency ratios *q_s_* shown in Figure 13.

Figure 14 shows the filtration characteristics of two cartridges whose filter bed was a composite of three factory-made E + B + A base layers (polyester + glass microfiber + cellulose).

A schematic diagram of the bed is shown in Figure 1. The filtration characteristics of both filter media show great similarity in terms of course and value. The test results for filters No. 1 and No. 2 have similar waveforms and values. Based on the test results of filter No. 1, we can see high initial filtration efficiency, which with a unit absorption *k_d_* = 2.84 g/m^2^ is at the level of *φ_0K_* = 99.87%. The initial filtration period is short because at *k_dpK_* = 5.95 g/m^2^ the filter achieves the established filtration efficiency of *φ* = 99.9%. During this time, the maximum diameters of dust grains in the air behind the filter do not exceed *d_pmax_* = 5–7 µm. When the filter reaches the dust absorption coefficient *k_dK_* = 17.4 g/m^2^, the size of the maximum dust grains stabilizes at the level *d_pmax_* = 2–3 µm. This accuracy is maintained until the filter reaches dust absorption *k_dK_* = 161.3 g/m^2^. This proves the very high accuracy of the tested deposit as a fibrous material that can be used in the automotive industry. The exceptionally high filtration efficiency and accuracy of both tested filters is maintained until the flow resistance Δ*p_f_* = 6.41 kPa, which significantly exceeds the value Δ*p_fdop_* = 4 kPa. Both tested filters achieved high dust absorption values when reaching Δ*p_fdop_* = 4 kPa, respectively *k_dK_* = 142.5 g/m^2^ filter no. 1 and *k_dK_* = 148.9 g/m^2^ filter no. 2. For the flow resistance Δ*p_f_* = 6.41 kPa dust absorption for filter no. 1 it is at the level of *k_dK_* = 195.9 g/m^2^. Such significant dust absorption values result mainly from the thickness of the composite bed (*g_m_* = 1.775 mm) and the configuration of the filter layers used. The first filtration layer (E) is a 0.62 mm thick polyester bed, which is characterized by high permeability and therefore low flow resistance. This is where depth filtration takes place and dust grains of the largest size are retained.

The second filtration layer is the B bed (glass macrofiber), which, due to the surface filtration process that takes place in it, has high separation efficiency and accuracy, with smaller grains being retained. The cellulose layer reinforces and stabilizes the airflow. Visually, one can see the absence of grains on its surface Figure 15c.

A comparative analysis of the filtration characteristics of the cartridge made of the triple-layer composite K (E + B + A) polyester–glass macrofiber–cellulose and the characteristics of the base materials is presented in Figure 16.

A clear difference in the course of the characteristics and the values of the parameters obtained between the tested materials A and B and K can be seen, which results mainly from the difference in the chemical composition of the filtration beds. The properties of filter beds A and B are discussed when analyzing the results shown in Figure 8. The three-layer composite K shows very good properties in terms of efficiency, filtration accuracy and pressure drop. The permissible value of pressure drops Δ*p_f_* = 4 kPa is achieved by composite K at an absorption coefficient of *k_dK_* = 142.5 g/m^2^, but operation of this bed with high efficiency and accuracy is possible up to the pressure drop Δ*p_f_* = 6.2 kPa at an absorption coefficient of *k_dK2_* = 195.9 g/m^2^. At this time, small dust grains in the range of *d_pmax_* = 1.5–3 µm are present in the air cleaned by the K-material filter. From the data presented it can be seen that, due to its high efficiency and high filtration accuracy, the composite material cartridge can be used up to a pressure drop of Δ*p_f_* = 6 kPa. Such permissible pressure drop values are used in truck and special vehicle air filters.

The systematic error Δ*φ_f_* of determining the filtration efficiency *φ_f_* of the filter cartridges for each measurement j from Equation (4) takes the following form:(10)$$∆\varphi_{f}=\left|\frac{\partial \varphi_{f}}{\partial_{F2}}{∆}_{mF2}\right|+\left|\frac{\partial \varphi_{f}}{\partial_{F1}}{∆}_{mF1}\right|+\left|\frac{\partial \varphi_{f}}{\partial_{D1}}{∆}_{mD1}\right|+\left|\frac{\partial \varphi_{f}}{\partial_{D2}}{∆}_{mD2}\right|$$
where the accuracy of the dust mass measurement is Δ*m_F2_ =* Δ*m_F1_ =* Δ*m_D1_ =* Δ*m_D2_ =* 0.0001 g.

The values of the errors in the determination of the filtration efficiency *φ_f_* of the filter cartridges made of different filter fabrics, tested at a constant filtration velocity *υ_F_* but for different values of the achieved dust absorption coefficient *k_d_*, assume different values within the range Δ*φ_f_* = 0.01–0.02%. Higher error values occur at lower values of the dust absorption coefficient kd, when the dust mass increments on the test cartridge are small due to shorter dust dosing periods.

By comparing the obtained systematic error values Δ*φ_j_* with the values of the obtained filtration efficiencies, it was found that the systematic errors were at least two orders of magnitude lower. In view of the above, it can be concluded that the systematic error values are so small that they do not affect the results obtained. The difference in the dust adsorption capacity of the various filter materials tested is statistically insignificant.

A comparative analysis of the performance of filters A, B, K, using the filtration quality factor *q_c_* and the filtration efficiency index *q_s_*, is shown in Figure 17.

The presented indicators clearly show that the composite filter K (cellulose–polyester–nanofiber) meets the required filtration efficiency, accuracy, and dust absorption.

Table 4 presents a summary of the parameters of the filter beds that are the results of tests and calculations.

It can be seen from the above summary that the three-layer filter bed (K-composite) made of single layers (polyester + glass microfiber + cellulose), for which the tests were performed under the same conditions as other materials, shows very good filtration properties. First, it has a high initial separation efficiency (99.87%), which translates into a short initial filtration period (*q_s_* = 4.21%) and high (*d_pmax_* = 1.5–3 µm) filtration accuracy over the entire basic working period. This makes the composite filter more than 60–250% higher than the other materials tested, with a dust absorption coefficient value of *k_dK_* = 142.5 g/m^2^.

Dust particles of different sizes were recorded in the air behind the filter, with particles of the smallest sizes (0.7 to 1.5 µm) being the most numerous, regardless of the type of filter material tested. Figure 18 shows the number of dust particles *N_p_* in successive measurement intervals in the air behind the filter elements tested for the first measurement.

In the air samples taken for analysis downstream of the filter, a decrease in the number of *N_p_* dust particles is observed as their dimension increases. In the last measurement interval, there is usually a single dust particle with a maximum size of *d_p_* = *d_pmax_*, which is a measure of the filtration accuracy.

For subsequent measurements, the number of dust particles decreases, as exemplified for filter material A (cellulose)—Figure 19.

In measurement no. 1 for material A, the maximum particle size was *d_pmax_* = 12 µm. In subsequent measurements, the number of particles decreases, and the dimension of the maximum particle decreases to *d_pmax_* = 3–4 µm. However, from measurement no. 10 onwards, the number of particles and the value of *d_pmax_* increases, which is consistent with the data in Figure 5. For the last measurement (no. 13), *d_pmax_* = 8 µm—Figure 19.

The numerical share of *U_p_* dust particles *N_p_* in the successive measurement intervals in the air behind filter element A (cellulose) and in the air entering the filter is shown in Figure 20.

The largest share of *U_p_* in the total number of dust particles in the air behind the filter is represented by dust particles with a size between *d_p_* = 0.7–1.5 µm. For filter element A (measurement no. 1), this share is *U_p_*_1_ = 31.3%. In subsequent measurements, the share of *U_p_* of these grains decreases and at measurement No. 5 (*k_d_* = 20.8 g/m^2^) is *U_p_*_5_ = 1%. Figure 20 shows (for comparison) the number shares of *U_p_* of the test dust particles in the range *d_p_* = 0.7–80 µm in the inlet air to the filter under test. The number shares of dust grains upstream and downstream of the filter are markedly different.

The proportion of test dust grains in the air in front of the filter from *d_p_* = 0.7 to *d_p_* = 1.5 µm is *U_p_* = 5.4–9.3%, and the proportion of dust grains of the same size behind the filter for the first measurement is more than 30%, indicating the high efficiency of large grains. The largest numerical share (*U_pmax_* = 16.2%) in the test dust is made up of dust grains with an equivalent diameter of dp = 4 µm. As the size of the grains increases, their share in the dust decreases and for 30 µm has a value of only *U_p_* = 0.01%.

## 5. Modeling of Vehicle Mileage

The time of proper operation of an air filter can be determined experimentally by performing laboratory tests or during road tests on a vehicle. However, these tests are costly and labor-intensive and require specialized apparatus. Similar results can be achieved using theoretical relationships available in the literature. According to the authors of the paper [60], the time of required operation of the air filter *τ_pf_* (reaching the permissible pressure drop ∆*p_fdop_*) is determined by the following relation:(11)$$\tau_{pf}=\frac{A_{c}\cdot k_{d}\cdot k_{c}}{Q_{Emax}\cdot s\cdot \varphi_{p}}\;\left[h\right]$$
where *A_c_* is the active surface area of the filter paper [m^2^]; *k_d_* is the absorption coefficient of the filter paper [g/m^2^] at the permissible value of pressure drop and flow resistance; *k_c_* is the coefficient taking into account the difference between the test parameters and the actual pollutants, i.e., the correction factor for the pollution parameters; *Q_Emax_* is the nominal air demand by the engine [m^3^/h], i.e., the nominal air flow rate; *s* is the average dust concentration of the air sucked into the filter [g/m^3^]; and *φ_p_* is the efficiency of the filter bed/matrix.

Assuming a constant (average) adopted driving speed *V_p_* (km/h), the distance traveled by the vehicle *S_p_* in time *τ_pf_* is expressed by the following relation:*S_p_* = *τ_pf_*·*V_p_* [km](12)

After taking into account relation (9), the distance traveled by the vehicle is determined by the following relation:(13)$$S_{p}=V_{p}\frac{A_{c}\cdot k_{d}\cdot k_{c}}{Q_{Emax}\cdot s\cdot \varphi_{p}}\;\left[\mathrm{km}\right]$$

The presented relationship shows that its use requires knowledge of the basic parameters of the filter material: filtration efficiency *φ_p_* = absorption coefficient *k_d_* for a specific value of permissible resistance. The kc coefficient is used to correct the effect of soot contained in polluted air during the use of the fibrous material. The value of the *k_c_* coefficient is assumed to be less than 1. In the case of paper filter beds which are elements of air filters in passenger cars, which are mainly used in urban conditions, the dominant component of air pollution sucked into engines is soot. Due to its properties, soot fills the filter bed more tightly than mineral dust, which causes a much faster increase in flow resistance and thus shortens the filter’s service life. For such filter operating conditions, a kc value less than 1 is assumed during theoretical analyses. In the case of air filter operation in conditions where the basic air pollutant is mineral dust (sandy areas), *k_c_* = 1 is assumed.

The effectiveness of modern filter materials used in automotive technology is known and, during the main period of filter operation, has values in the range of *φ_p_* = 99.5–99.9%. about the vehicle mileage, is the dust absorption coefficient kd. The research results in this study show that the filter materials differ quite significantly in the values of the kd coefficient, which was determined at the established permissible flow resistance Δ*p_fdop_* = 4 kPa. Using the relationship (11), the mileage of a passenger car was modeled for the kd coefficient values for cellulose and K-composite. The remaining parameters in relation (11) were assumed as constant.

It was assumed that the vehicle would be used mainly in non-urban conditions, where the main component of air pollution is mineral dust. It was assumed that the average speed of the moving vehicle is *V_p_* = 60 km/h, and the dust concentration in the air surrounding the vehicle is *s* = 0.001 g/m^3^ [25]. The mileage of an Audi A4 vehicle was modeled, the power unit of which is a diesel engine with turbocharging and air cooler, with a displacement of *V_ss_* = 2.496 dm^3^. The air demand of the engine operating at rated conditions is *Q_Emax_* = 554 m^3^/h. The air filter is equipped with a filter insert with an active material area of *A_c_* = 2.09 m^2^. The value of the coefficient kd and the value of filtration efficiency *φ_p_* of filters A, B, C and K were adopted from the tests presented in Figure 7, Figure 12 and Figure 16.

The calculation results of the vehicle mileage at which the filter will be serviced due to the achievement of the permissible flow resistance Δ*p_fdop_* are shown in Figure 21.

A higher value of the dust absorption coefficient kd gives a greater vehicle mileage, which is as expected. An air filter using composite filter material K (E + B + A) provides twice (33,700 km) greater mileage to achieve Δ*p_fdop_* than other filter materials. This value may change for other conditions of use. For example, changing the dust concentration from 0.001 to 0.002 g/m^3^ shortens the vehicle’s mileage by 50% (Figure 22).

## 6. Conclusions

Experimental studies were conducted to evaluate the filtration properties of various materials in order to determine the optimal material for filtering the air drawn in by machine drive engines. This evaluation particularly considered the inclusion of a new composite bed, focusing on key filtration characteristics such as separation efficiency and accuracy, as well as pressure drop, all of which were measured on a test bench. The materials tested included factory-standard filter options like cellulose, polyester, and micro glass fiber, alongside a specially designed filter bed that integrated three layers of distinct materials. This innovative filter bed was constructed from three foundational materials: polyester, glass microfiber, and cellulose. Its filtration characteristics were meticulously evaluated to gauge performance. The analysis extended to compare the air filtration quality coefficients derived from the fiber baffles, offering a detailed comparative perspective based on empirical data.

An important aspect of the present work is that it was an experimental study carried out on a laboratory test bench using real filter materials and test dust replicating the real dust contaminating the air drawn into internal combustion engines. The bench’s capabilities allowed the testing of pleated filter materials in a dust concentration range of up to 2 g/m^3^ and air flow rates up to 80 m^3^/h. Another important aspect of the work is that a unique test methodology was been used to perform material characterization, with a small surface area *A_c_* = 0.183 m^2^, which significantly reduced the testing time and costs. In this respect, it was assumed that the filtration parameters obtained from the tests would be equivalent for filters of the same material with a larger material area but the same test conditions, including dust concentration and filtration velocity.

The comprehensive analysis and results obtained from this study culminated in insightful conclusions regarding the efficacy of different filter materials and configurations. These conclusions are poised to contribute significantly to the selection and optimization of filter materials for enhancing the performance and longevity of internal combustion engines.

Experimental studies using test dust have shown that the initial filtration period, the length of which varies depending on the type of fiber material used, has been defined as the filter operating time required to achieve the target separation efficiency of *φ* = 99.9%. This is crucial for the evaluation of filter materials, in terms of their ability to quickly achieve a high level of separation efficiency for particles, especially those above 5 µm in size.The initial phase of filtration is marked by a low yet gradually increasing efficiency *φ_f_* and filtration accuracy *d_pmax_*, as well as small but increasing pressure drop Δ*p_f_*. During this period, larger dust particles were detected in the air downstream of the filters being examined, with their sizes diminishing progressively as the filtration process continued. The largest size of dust grains (*d_pmax_* = 12 µm) was registered in the air behind filter A (cellulose), and the smallest, *d_pmax_* = 7 µm, behind filter B (glass microfiber) and *d_pmax_* = 5 µm behind filter with composite bed K (polyester–glass microfiber–cellulose).The initial period of filtration, characterized by the presence of dust grains larger than 5 µm, which poses a significant risk to the operation of internal combustion engines, causing accelerated wear of its key components: piston, piston rings and cylinder. In the case of filter material A (cellulose), the initial period exceeds 22% of the total operating time of the filter cartridge. This underscores the need to supplement cellulose with other materials in filter beds to mitigate the drawbacks of its initial filtration efficiency.When a new filter element made of fibrous material is installed in a vehicle filter, it begins to provide the required separation efficiency and accuracy only after the vehicle has driven 4–30% of its expected mileage. This delay before achieving the required filtration efficiency (99.9%) can lead to frequent and premature filter element replacements. Such replacements entail unnecessary costs, but also the risk of faster wear of engine components. Therefore, knowing the filtration properties of the materials and the service life of the filter elements is critical to optimizing service schedules and protecting the engine from premature wear.The filter bed (K-composite) made for this study, in the form of three composite layers of polyester–glass microfiber–cellulose materials, shows, under the same test conditions as other materials, a high initial separation efficiency (99.98%), a short initial filtration period (*q_s_* = 4.21%), high (*d_pmax_* = 1.5–3 µm) accuracy over the entire basic period of filter operation and more than 60–250% higher value of absorption coefficient (*k_dK_* = 142.5 g/m^2^).

The high dust absorption capacity of the K-composite (polyester–glass macrofiber–cellulose) will ensure longer (on average 2 times) vehicle mileage until air filter servicing is performed—filter cartridge replacement. The mileage of the car may reach different values in operation than modeled, which is influenced by the conditions of use and, above all, the concentration of dust in the air.

The results of the characteristics of the filter materials obtained during the experimental studies are partly part of the information gap in the basic properties of filter materials that are used in the construction of filter cartridges for the intake air of internal combustion engines of motor vehicles.

The results obtained will allow the proper selection of filter materials for the intake air of a motor vehicle’s internal combustion engine, so as to minimize engine wear and extend vehicle mileage as much as possible.

## Figures and Tables

**Figure 1 materials-17-03249-f001:**
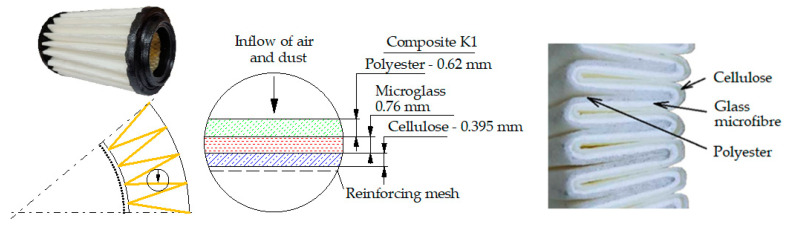
Filter layers in multilayer bed (composite) K (polyester + glass microfiber + cellulose).

**Figure 2 materials-17-03249-f002:**
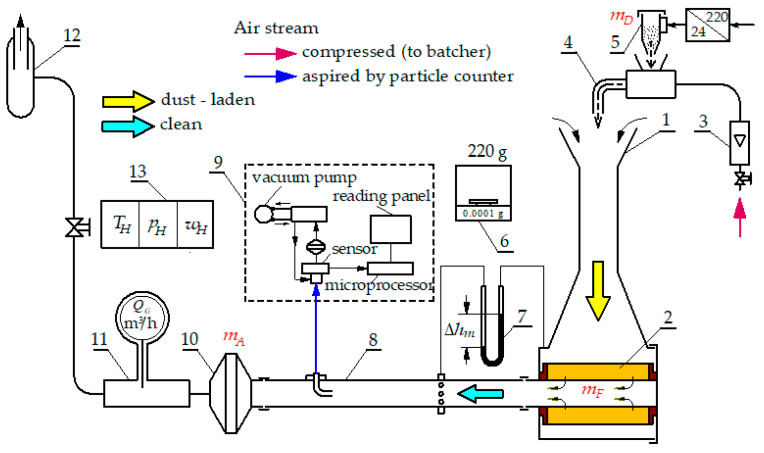
Functional diagram of the station: 1—dust mixing chamber, 2—tested cartridge, 3—“rotameter” type flowmeter, 4—dust dosing device, 5—dosing dust container, 6—scale for measuring the mass of filters, 7—U-tube liquid manometer, 8—measuring pipeline, 9—particle counting device, 10–measuring (protective) filter, 11—air flow meter, 12—suction fan, 13—environmental parameters control device.

**Figure 3 materials-17-03249-f003:**
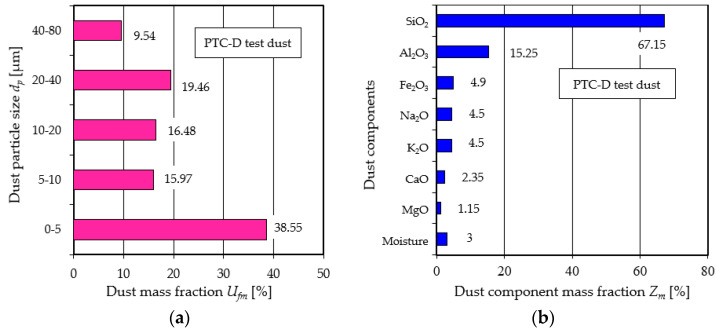
Characteristics of PTC-D test dust: (**a**) mass proportion of specific fractions in the dust, (**b**) mass proportion of individual components in the dust [53] Publisher Eksploat. I Niezawodn.–Maint. Reliab. 2015.

**Figure 4 materials-17-03249-f004:**
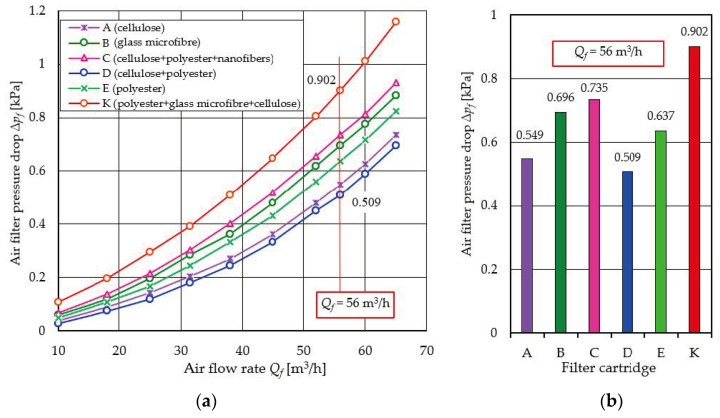
Flow resistance of filter media: (**a**) characteristics Δ*p_f_* = *f*(*Q_f_*), (**b**) flow resistance for *Q_f_* = 56 m^3^/h.

**Figure 5 materials-17-03249-f005:**
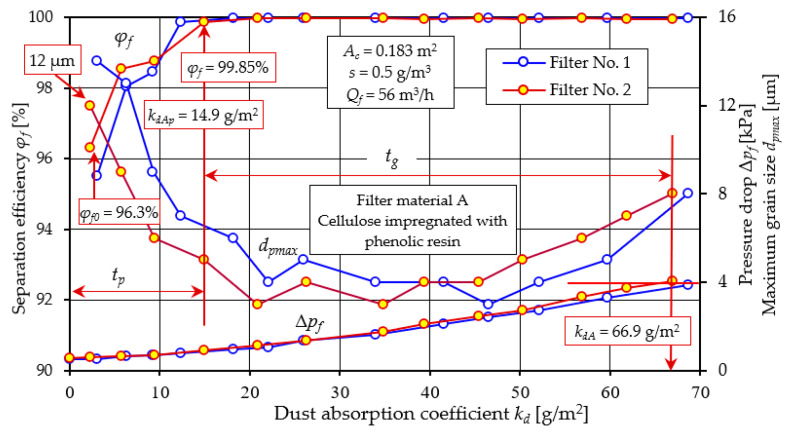
Characteristics of efficiency *φ_f_*, filtration accuracy *d_pmax_*, and pressure drop Δ*p_f_ = f*(*k_d_*) as a function of the dust absorption coefficient *k_d_* of the tested cartridges with filter material A (cellulose).

**Figure 6 materials-17-03249-f006:**
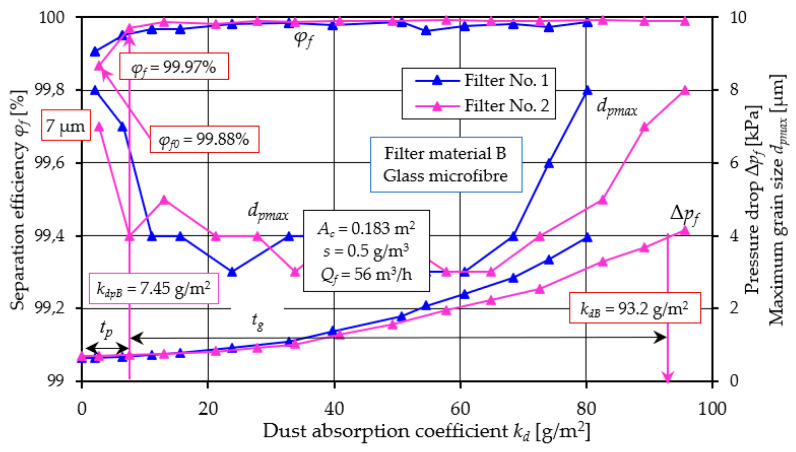
Characteristics *φ_f_ = f*(*k_d_*), *d_pmax_* = *f*(*k_d_*), and Δ*p_f_ = f*(*k_d_*), depending on the dust absorption coefficient *k_d_* of the tested filter B (glass microfiber).

**Figure 7 materials-17-03249-f007:**
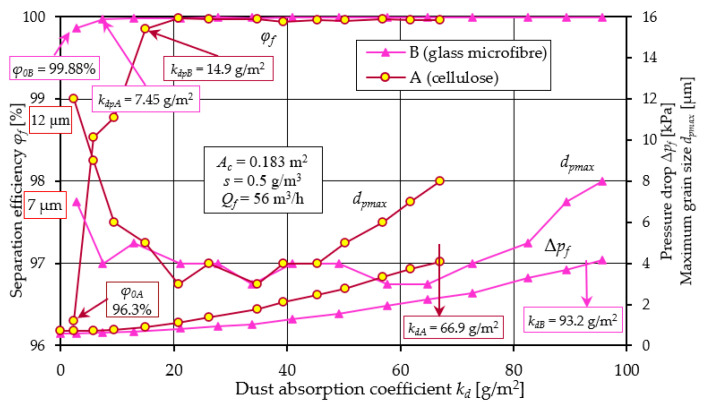
Comparative analysis of the characteristics of *φ_f_ = f*(*k_d_*), *d_pmax_* = *f*(*k_d_*), and Δ*p_f_ = f*(*k_d_*), depending on the dust absorption coefficient kd of the tested filters A (cellulose) and B (glass microfiber).

**Figure 8 materials-17-03249-f008:**
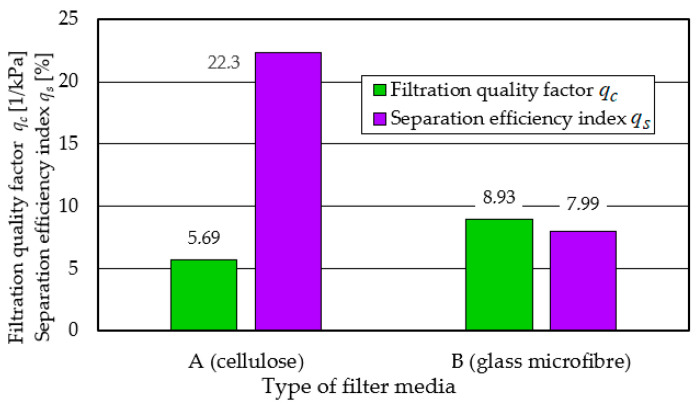
Comparative analysis of the quality factor *q_c_* and *q_s_* index of the tested filter inserts: A (cellulose) and B (glass macrofiber).

**Figure 9 materials-17-03249-f009:**
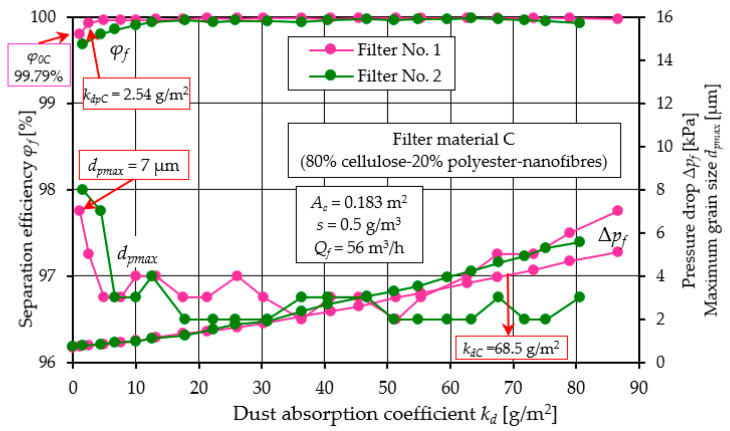
Characteristics *φ_f_ = f*(*k_d_*) and *d_pmax_* = *f*(*k_d_*) and Δ*p_f_ = f*(*k_d_*) depending on the dust absorption coefficient *k_d_* of the tested filters: C (cellulose–polyester–nanofibers).

**Figure 10 materials-17-03249-f010:**
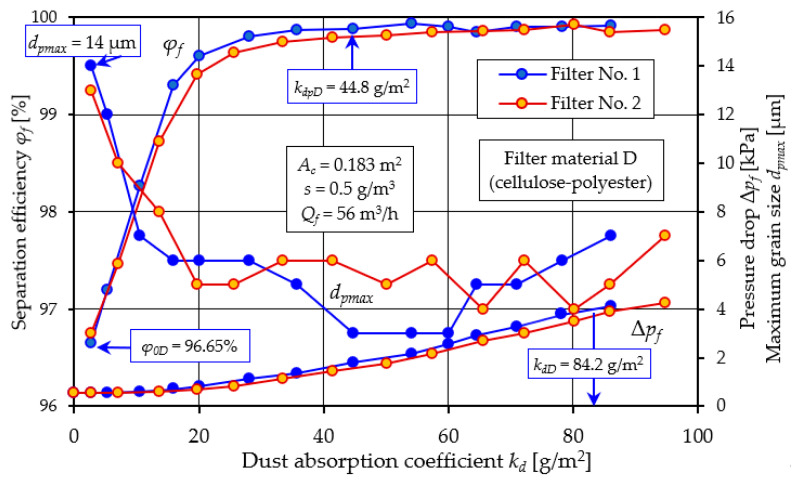
Characteristics *φ_f_ = f*(*k_d_*) and *d_pmax_* = *f*(*k_d_*) and Δ*p_f_ = f*(*k_d_*) depending on the dust absorption coefficient *k_d_* of the tested filters: D (cellulose–polyester).

**Figure 11 materials-17-03249-f011:**
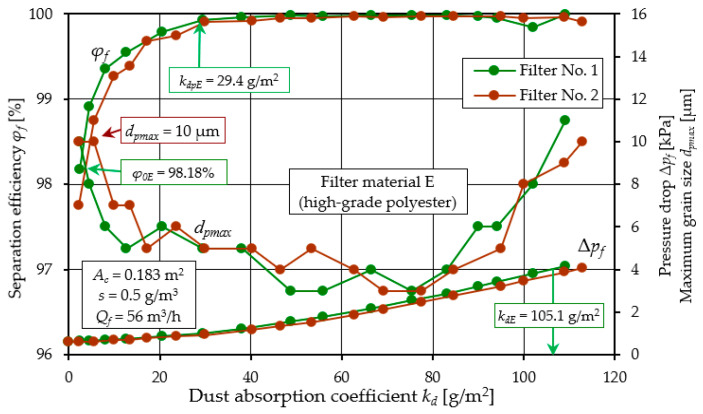
Characteristics *φ_f_ = f*(*k_d_*) and *d_pmax_* = *f*(*k_d_*) and Δ*p_f_ = f*(*k_d_*) depending on the dust absorption coefficient *k_d_* of the tested filters: E (polyester).

**Figure 12 materials-17-03249-f012:**
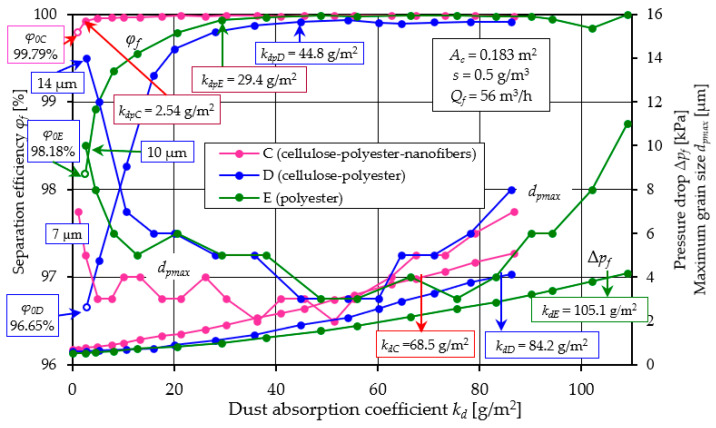
Characteristics *φ_f_ = f*(*k_d_*) and *d_pmax_* = *f*(*k_d_*) and Δ*p_f_ = f*(*k_d_*) depending on the dust absorption coefficient *k_d_* of the tested filters: C (80% cellulose-20% polyester–nanofibers, D (cellulose–polyester) and E (polyester).

**Figure 13 materials-17-03249-f013:**
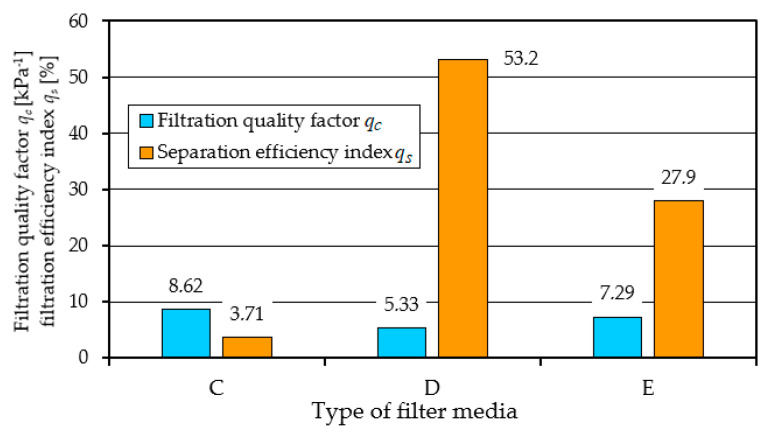
Comparative analysis of the quality factor *q_c_* and *q_s_* index of the tested filter inserts: C (cellulose–polyester–nanofibers, D (cellulose–polyester), and E (polyester).

**Figure 14 materials-17-03249-f014:**
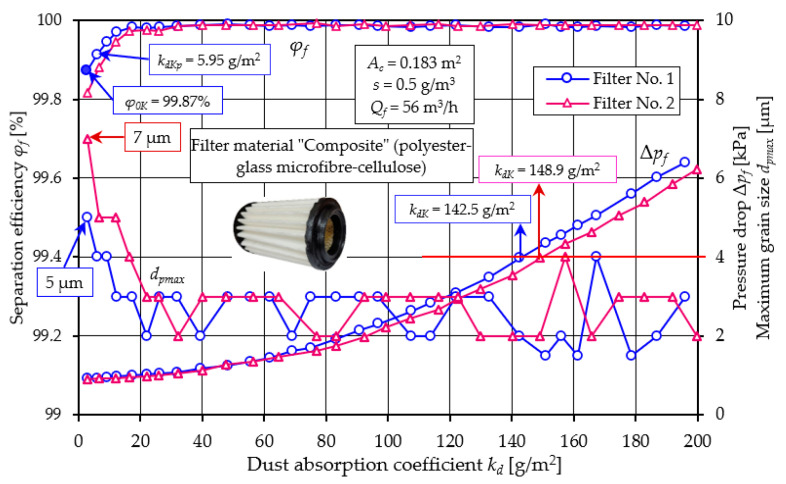
Characteristics *φ_f_ = f*(*k_d_*) and *d_pmax_* = *f*(*k_d_*) and Δ*p_f_ = f*(*k_d_*) as a function of the dust absorption coefficient kd of a K cartridge with a filter bed consisting of three layers of basic materials: E + B + A.

**Figure 15 materials-17-03249-f015:**
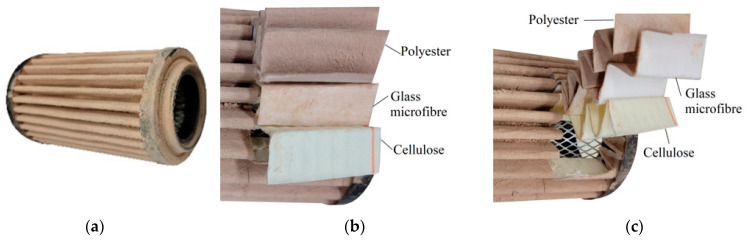
K (E + B + A) composite bed filter after dust testing: (**a**) view of the filter, (**b**) inlet sides, and (**c**) outlet sides of successive filter layers.

**Figure 16 materials-17-03249-f016:**
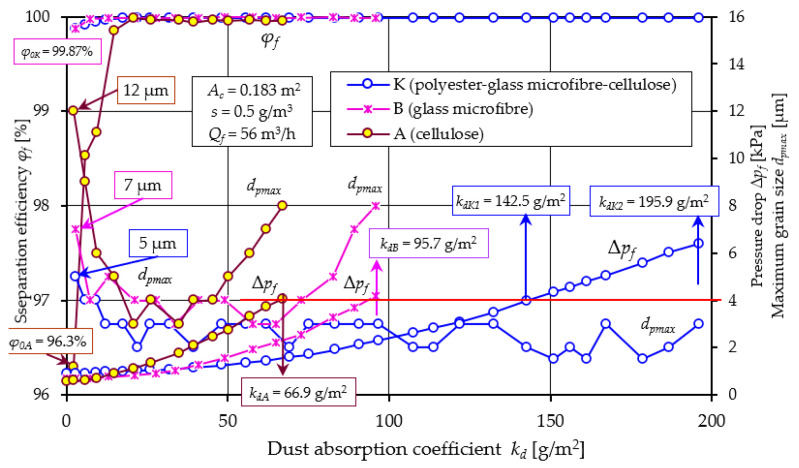
Characteristics of *φ_f_ = f*(*k_d_*), *d_pmax_* = *f*(*k_d_*), and Δ*p_f_ = f*(*k_d_*) as a function of the dust absorption coefficient kd of a bed consisting of three layers—K (E + B + A)—and base beds B (glass macrofiber) and A (cellulose).

**Figure 17 materials-17-03249-f017:**
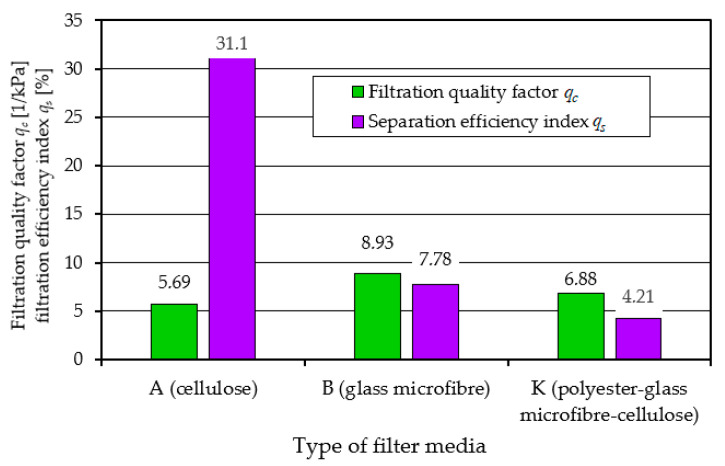
Filtration quality factor *q_c_* and separation efficiency index *q_s_* of the tested filter cartridges: A (cellulose), B (glass macrofiber) and composite K (polyester–glass macrofiber–cellulose).

**Figure 18 materials-17-03249-f018:**
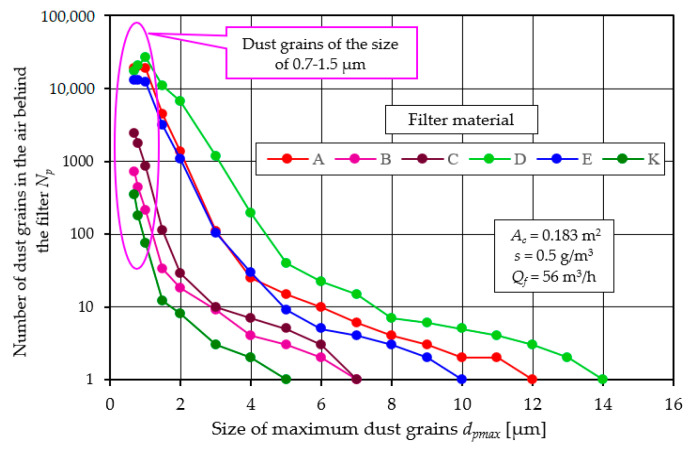
Number of dust grains *N_p_* during the first measurement in successive measurement intervals in the air behind the filter elements tested.

**Figure 19 materials-17-03249-f019:**
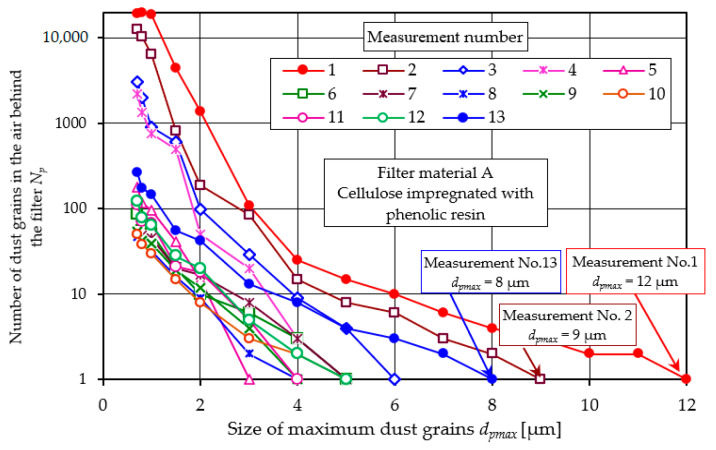
The number of *N_p_* dust grains present in the air cleaned by filter element A (cellulose bed) during successive measurements (1–13).

**Figure 20 materials-17-03249-f020:**
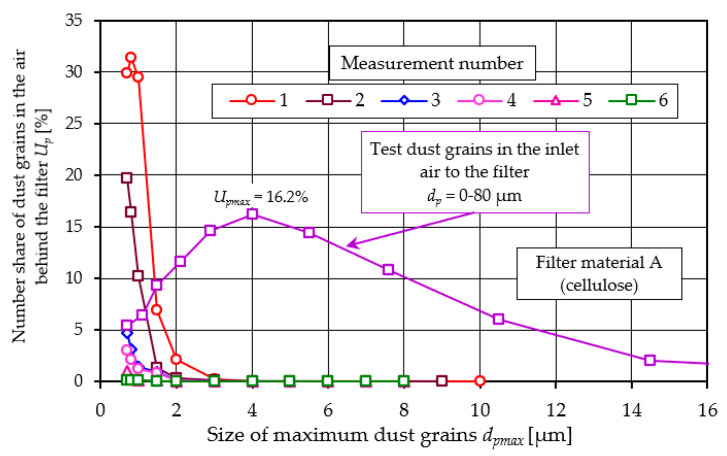
Number contribution of *U_p_* particles *N_p_* in successive measurement intervals in the air downstream of filter element A (cellulose) and in the air entering the filter.

**Figure 21 materials-17-03249-f021:**
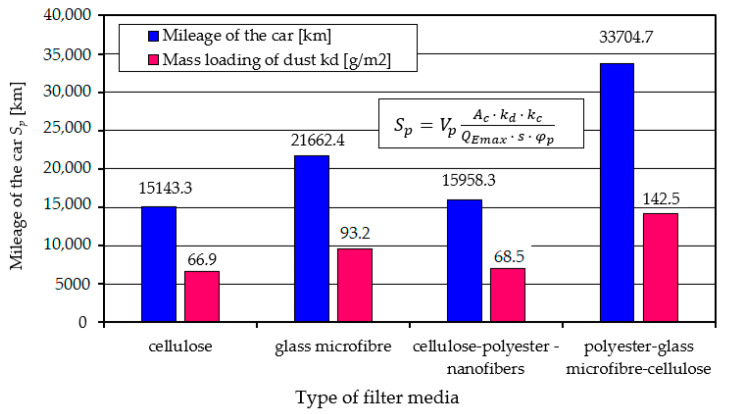
Modeled car runs for different dust absorption coefficients *k_d_* of filter beds.

**Figure 22 materials-17-03249-f022:**
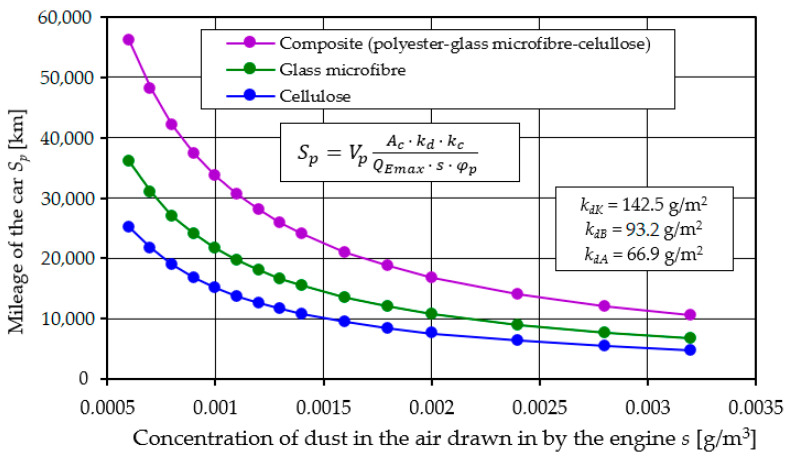
The course of the car as a function of dust concentration in the air for the dust absorption coefficients *k_d_* of the tested filter materials.

**Table 1 materials-17-03249-t001:** Dust concentrations in various air conditions [28].

Environmental Conditions	Value [g/m^3^]
Clean village environments	from 0.00001
Tracked vehicle column movement in desert conditions	about 20
Dusty environments	0.001–10
A few meters from the sandy road on which all-terrain vehicles are moving	0.05–10
On highways	0.0004–0.1
Driving vehicle columns on a sandy terrain	0.03–8
During helicopter take-off or landing on an accidental landing site at the height of the helicopter’s propeller end, i.e., 0.5 m above the ground	3.33
At the inlet of the intake system of a vehicle’s internal combustion engine	no more than 2.5
Limited visibility	0.6–0.7
Zero visibility	about 1.5

**Table 2 materials-17-03249-t002:** Parameters of filter materials.

Contractual Designation	Filter Paper Identification	Filtration Material	Permeability *q_pb_* [dm^3^/m^2^/s]	Grammage *g_m_* [g/m^2^]	Thickness *g_z_* [mm]	Max. Pore Size *d_ps_* [µm]
A	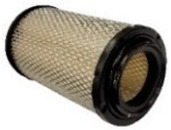	Cellulose impregnated with flame-retardant phenolic resin	255	130	0.395	55
B	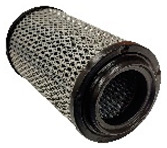	Glass microfiber	190	-	0.76	35
C	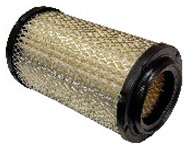	80% cellulose + 20% polyester + nanofibers Acrylic impregnation Nanofiber coating Flame-retardant	139	130	0.32	45
D	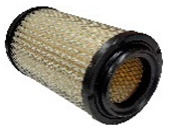	80% cellulose + 20% polyester High hydrophobicity High tensile strength	235	135	0.45	-
E	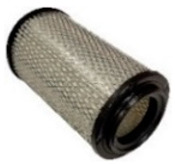	Polyester High hydrophobicity. High tensile strength	136	260	0.54	-
K (E-B-A)	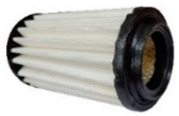	Polyester + glass microfiber + cellulose	-	-	1.775	-

**Table 3 materials-17-03249-t003:** SEM images (500× magnification) of the inlet and outlet sides of the samples before testing.

Name of the Material	Inlet Side	Outlet Side
A (cellulose),	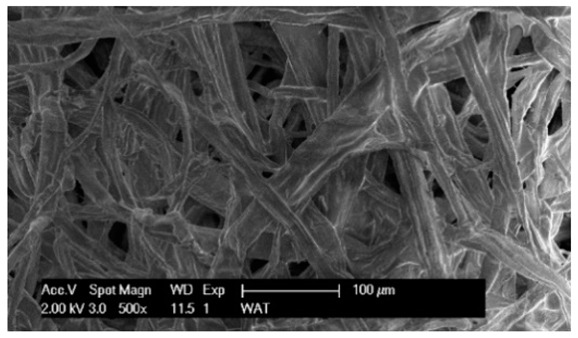	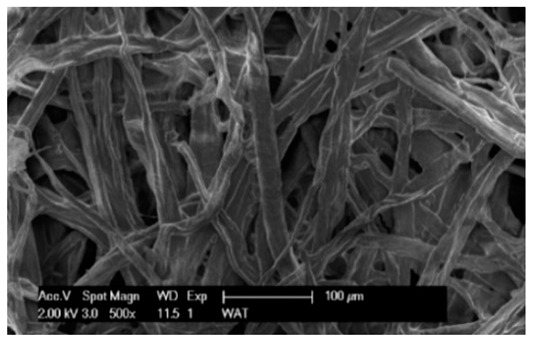
B (glass microfiber)	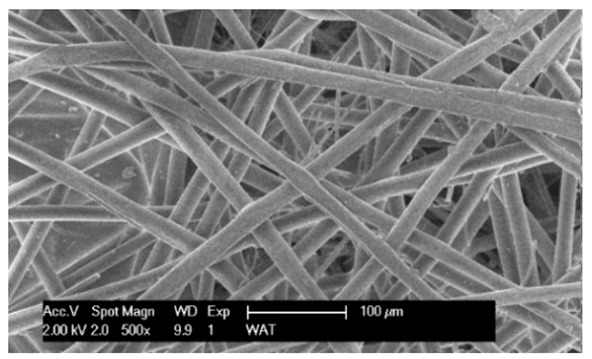	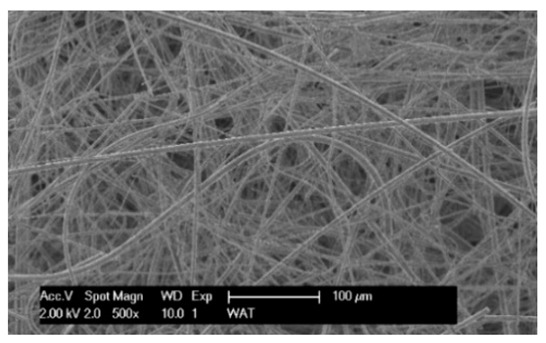
C (cellulose–polyester–nanofibers)	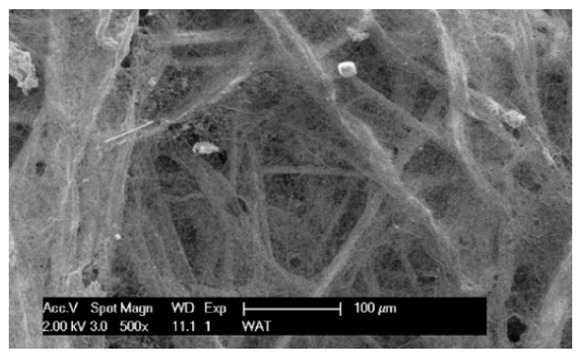	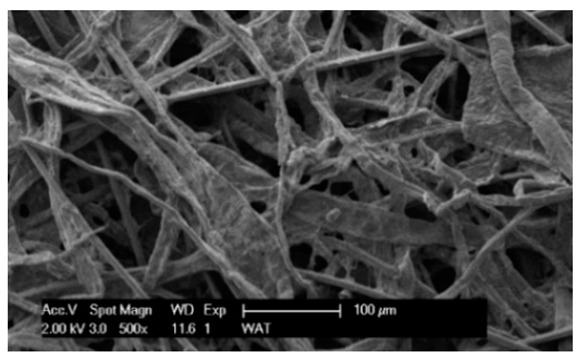
D (cellulose–polyester)	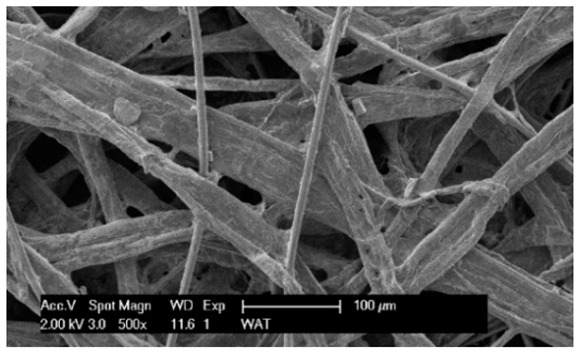	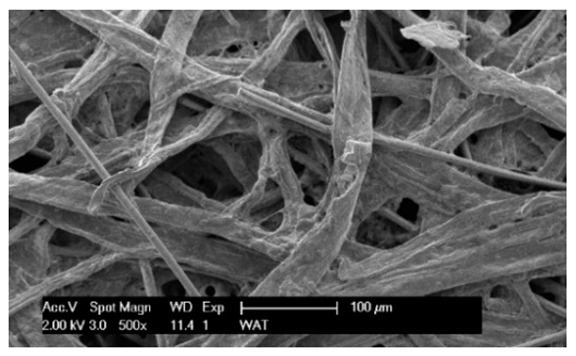
E (polyester)	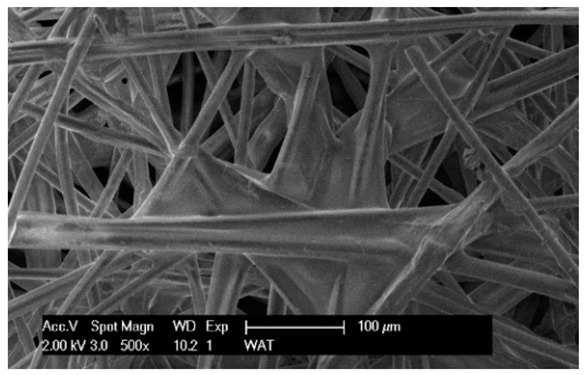	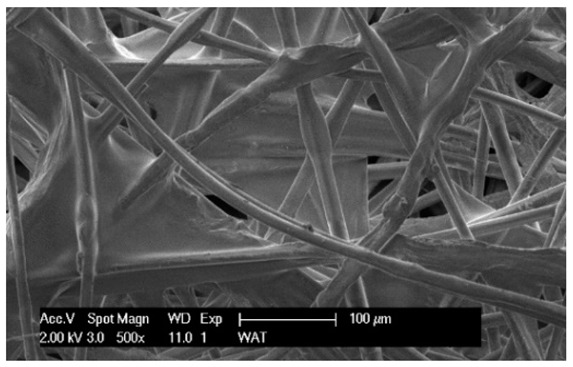

**Table 4 materials-17-03249-t004:** Summary of parameters of filter beds that are the results of tests and calculations.

Filter	Filter Bed	*φ*_0_ [%]	*d_pmax_* [µm]	*k_dp_* [g/m^2^]	*k_d_* [g/m^2^]	*q_s_* [%]	*q_c_* [kPa^−1^]
A	Cellulose	96.31	12	14.9	66.9	22,3	5.69
B	Glass microfiber	99.88	7	7.45	93.2	7,99	8.93
C	Cellulose–polyester–nanofibers	99.79	7	2.54	68.5	3.71	8.62
D	Cellulose–polyester	96.65	14	44.8	84.2	53.2	5.33
E	Polyester	98.18	10	29.4	105.1	27.9	7.29
K (E-B-A)	Polyester + glass microfiber + cellulose	99.87	5	5.95	142.5	4.21	6.88

## Data Availability

Data sharing is not applicable to this article.

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
