# Peer review of "Experimental Dust Absorption Study in Automotive Engine Inlet Air Filter Materials"

_materials, 2024, doi:10.3390/ma17133249_

Round 1
Reviewer 1 Report
Comments and Suggestions for Authors
Based on my expertise, this article is of high quality, both for the results reached and for their description/illustration/discussion. Therefore, I suggest to accept It, after reviewing the text according to these minor comments.
Lines 69-73: I suggest to use the same units for all the data.
Line 219: Revise the apex.
Equation 9 KPa^-1 would be better.
The format of figures should be revised. Use Palatino Linotype.
The last two paragraphs of Conclusions should not have a numeration; remove 7 and 8.
Author Response
Answers to Reviewer v1
Experimental Study of Dust Absorption of Automotive Engine Inlet Air Filter Materials
Tadeusz Dziubak
Reviewers' comments:
Comments and Suggestions for Authors
Based on my expertise, this article is of high quality, both for the results reached and for their description/illustration/discussion. Therefore, I suggest to accept It, after reviewing the text according to these minor comments.
- Lines 69-73: I suggest to use the same units for all the data.
Answer:
The units were standardised "1x10-5 g/m3".
- Line 219: Revise the apex.
Answer:
Corrected "3100 l/(m2/s)".
- Equation 9 KPa^-1 would be better.
Answer:
The unit notation in the equation has been corrected to the following "kPa-1".
- The format of figures should be revised. Use Palatino Linotype.
Answer:
As suggested, the format of the drawings was changed and the Palatino Linotype font was used.
- The last two paragraphs of Conclusions should not have a numeration; remove 7 and 8.
Answer:
As suggested, the last two points (7 and 8) of the proposals are not numbered.
Many thanks for your insightful and informative review
Reviewer 2 Report
Comments and Suggestions for Authors
This paper presents a substantial amount of experimental and calculation results. The presentation of results are not well-organized and some suggestions are given below.
1. There are too many symbols and definitions utilized by this study. A nomenclature section is suggested. It helps readers to follow.
2. The values of qs and qc of all filters are summarized in Table 2. Therefore, Figures 8, 13, and 17 are redundant and they can be omitted.
3. Conclusion is too lengthy to catch the major findings and contribution of this work.
4. There are some inconsistencies. In #5 of conclusion, high (dpmax = 1.5-3 μm) accuracy and absorption coefficient (kdK = 142.5 g/m2) are given for the filter K. However, dpmax = 5 μm is listed in Table 2 and kd is used in Table 2 rather than kdK.
5. In terms of efficiencies qs and qc, the composite filter K is not the best choice. Authors are suggested to explain why the filter K can be considered as the promising material.
6. The filter K is a composite of E, B, and A. What are the physical mechanisms causing the significant decrease of qs for the filter K as compared to the filters E, B, and A? Also, why is qc of the filter K lower than those of the filters E and B?
Author Response
Answers to Reviewer 2 v1
Experimental Study of Dust Absorption of Automotive Engine Inlet Air Filter Materials
Tadeusz Dziubak
Reviewers' comments:
Comments and Suggestions for Authors
This paper presents a substantial amount of experimental and calculation results. The presentation of results are not well-organized and some suggestions are given below.
- There are too many symbols and definitions utilized by this study. A nomenclature section is suggested. It helps readers to follow.
Answer:
As suggested, nomenclature was done and included in the paper.
- The values of qs and qc of all filters are summarized in Table 2. Therefore, Figures 8, 13, and 17 are redundant and they can be omitted.
Answer:
The reviewer's suggestion is obvious but presenting the values of qs and qc graphically (Figures 8, 13 and 17) after successive tests of the filter group will help readers to better follow the results of the study. There is no need to look for the data in Table 2, which is a summary of all the results and can be found at the end of the results discussion. Therefore, the author believes that (Figures 8, 13 and 17) should remain in their current form.
- Conclusion is too lengthy to catch the major findings and contribution of this work.
Answer:
The content of the proposals has been revised and shortened and included in the work.
- There are some inconsistencies. In #5 of conclusion, high (dpmax = 1.5-3 μm) accuracy and absorption coefficient (kdK = 142.5 g/m2) are given for the filter K. However, dpmax = 5 μm is listed in Table 2 and kd is used in Table 2 rather than kdK.
Answer:
Table 2 shows the filtration accuracy of dpmax = 5 μm. This is the maximum dust grain size after the first measurement (Fig. 16). The value of this grain is always larger at the beginning and later in the baseline period the size decreases. In application 5, the filtration accuracy (dpmax = 1.5-3 μm) is given, which refers to the process during the basic period of the filter.
The designation of the coefficient “kd” in the title of Table 2 applies to all the filters tested. However, in the figure descriptions, the filter designation is added to the “kd” designation: for example, for filter A it is “kdA”, for filter K it is “kdK”.
- In terms of efficiencies qs and qc, the composite filter K is not the best choice. Authors are suggested to explain why the filter K can be considered as the promising material.
Answer:
The K composite filter is distinguished by such positive features as high initial efficiency (99.87%) and filtration accuracy (dpmax = 5 µm), as well as high dust absorption (kdK = 142.5 g/m2), which exceeds the absorption of other tested materials by 100%. These factors ensure that the air supplied to the engines will have a sufficiently high purity, which will affect less wear and tear on engine components and increase its service life. In addition, it should be noted that the filtration accuracy of the K-composite remains at 2-3 µm for almost the entire period of the filter's operation, which was not recorded during the testing of other materials. Filters for inlet air of internal combustion engines are required to have filtration accuracy of dust grains above 3 µm. The high dust absorption capacity of the K-filter will allow the filter cartridge replacement period to be doubled, which will reduce operating costs. A vehicle performance test using the K composite filter is planned.
- The filter K is a composite of E, B, and A. What are the physical mechanisms causing the significant decrease of qs for the filter K as compared to the filters E, B, and A?
Answer:
The qs factor determines the percentage of the initial period tp (obtaining the kdp factor) in the total period of filtration, which ends with reaching the permissible resistance and obtaining the dust absorption factor kdF. The better filtration properties are those of the material whose duration of the initial period of the filtration process will be shorter, that is, it will reach the established filtration efficiency (99.9%) sooner. Material E, which is the first filtration layer in the K-composite, has a significant duration of the initial period, and its kdp = 29.4 g/m2. In this layer, depth filtration occurs and dust grains of the largest size are retained. Dust grains of small sizes are directed to material B, which is the second filtration layer in the composite. The duration of the initial period of this layer is determined by the factor kdp = 7.45 g/m2. The purpose of this layer, on which surface filtration takes place, is to trap small-sized dust grains. The filtration process in this composite occurs on two layers in the form of two-stage filtration, and layer A stabilizes the flow. These phenomena result in a much shorter time for the K-composite to reach the set filtration efficiency (99.9%), and the ratio qs.
Also, why is qc of the filter K lower than those of the filters E and B?
Answer:
The filtration quality factor qc refers to the initial filtration efficiency and the corresponding pressure drop of the same filter material. Both parameters determine the value of the qc factor. Lower efficiencies or higher flow resistances cause the value of qc to decrease. For filter K, the value of qc = 6.88 kPa-1 and is lower than the coefficient for filter B (qc = 8.93 kPa-1), which is mainly due to the significantly higher (by more than 28%) initial flow resistance of filter K (Fig. 4). The efficiency values of both filters are at the same level of 99.8%. For filter E, the value of qc = 7.29 kPa-1 and is lower than filter B due to the lower initial filtration efficiency (99.18%), but higher than filter K due to the much higher flow resistance of filter K.
Many thanks for your insightful and informative review
Reviewer 3 Report
Comments and Suggestions for Authors
Comments:
1. Abstract: At the end, add a brief statement on the important implications of this work.
2. In the first paragraph of the introduction, could you mention what are the major pollutants and their typical concentrations in the dust/aerosols generated from road surfaces.
3. Line 288: “The filter beds (A, B, C, D, E) are factory-made products…” Are they purchased from a commercial company?
4. Authors are suggested to do some statistical analysis to show whether the difference in the dust adsorption capacity in different filter materials are statistically significant.
5. Before Conclusions, add a section describing the important implications and limitations of this work
6. Conclusions: It is too long and not focused. Please condense it by highlighting the key results and observations obtained in this study.
7. Figure 3: Explain the abbreviation “PTC-D” in the figure legend.
8. Figure 8: Could you provide more information about why the separation efficiency index of cellulose material is considerably higher than glass microfibre.
Comments on the Quality of English LanguageMinor English language is suggested at this stage.
Author Response
Answers to Reviewer 3 v1
Experimental Study of Dust Absorption of Automotive Engine Inlet Air Filter Materials
Tadeusz Dziubak
Comments and Suggestions for Authors
Comments:
- Abstract: At the end, add a brief statement on the important implications of this work.
Answer:
The summary was supplemented with the following text:
A filtration composite bed constructed from a clique of materials that differ in filtration parameters can be, due to its high filtration efficiency and accuracy and high dust absorption, an excellent filter material for engine intake air. The filtration parameters of the composite will depend on the type of filter layers and the order in which they are arranged in relation to the aerosol flow. This paper presents a methodology for the selection and testing of various filter materials.
- In the first paragraph of the introduction, could you mention what are the major pollutants and their typical concentrations in the dust/aerosols generated from road surfaces.
Answer:
The introduction was supplemented with information regarding the concentration values of road dust and dust from brake and tire wear.
- Line 288: “The filter beds (A, B, C, D, E) are factory-made products…” Are they purchased from a commercial company?
Answer:
The filter beds were purchased from a commercial company, hence the contractual designation (A, B, C, D, E). The company is entitled to the detailed test results. The data that support the research results will be available after an embargo from the date of publication to allow commercialisation of the research results.
- Authors are suggested to do some statistical analysis to show whether the difference in the dust adsorption capacity in different filter materials are statistically significant.
Answer:
A statistical analysis was carried out and it was found that the systematic errors are at least two orders of magnitude smaller than the obtained values of filtration efficiency. In view of the above, it can be concluded that the values of the systematic errors are so small that they do not affect the results obtained. The content of the work has been supplemented with relevant information in this regard.
- Before Conclusions, add a section describing the important implications and limitations of this work
Answer:
As suggested, the following section was added:
An important aspect of the present work is that it is an experimental study carried out on a laboratory test bench using real filter materials and using test dust replicating the real dust with which the air drawn into internal combustion engines is contaminated. The bench's capabilities allow testing of pleated filter materials in the dust concentration range of up to 2 g/m3 and for air flow rates of up to 80 m3/h. Another important aspect of the work is that a unique test methodology has been used to perform material characterizations with a small surface area Ac = 0.183 m2, which significantly reduces testing time and costs. In this respect, it was assumed that the filtration parameters obtained from the tests would be equivalent for filters with a larger material area of the same material, but with the same test conditions, such as dust concentration and filtration velocity.
- Conclusions: It is too long and not focused. Please condense it by highlighting the key results and observations obtained in this study.
Answer:
The content of the proposals has been revised and shortened and included in the work.
- Figure 3: Explain the abbreviation “PTC-D” in the figure legend.
Answer:
The abbreviation "PTC-D" appearing in Figure 3 denotes the test dust, which was produced in Poland and its chemical and granulometric composition corresponds to the "AC fine" test dust, which is used in many test centres around the world. The designation "PTC-D" comes from the Polish words and means: "fine test dust".
- Figure 8: Could you provide more information about why the separation efficiency index of cellulose material is considerably higher than glass microfibre.
Answer:
The factor qs determines the percentage of the initial period in the total filtration period determined by the permittivity, to which the dust absorption coefficient kdF corresponds. If two filter materials with the same value of kdF have the same value of kdp, the one with the better filtration properties, which has a lower value of kdp, will reach the set filtration efficiency sooner. The duration of the initial period of the filtration process will be shorter.
In the case of material A (cellulose) and B (glass microfibre), cellulose has a longer initial period due to its higher permeability and smaller bed thickness than the material in filter B. Thus, the qs factor has a higher value.
Comments on the Quality of English Language
Minor English language is suggested at this stage.
Answer:
As suggested, a correction was made to the English language
Many thanks for your insightful and informative review
Round 2
Reviewer 2 Report
Comments and Suggestions for Authors
The revised manuscript properly replied the reviewer's comments and it can be accepted for publication.
Reviewer 3 Report
Comments and Suggestions for Authors
The quality of the manuscript is improved after revision. The revised version can be accepted.
Comments on the Quality of English LanguageMinor editing is suggested.